# Let-7a-regulated translational readthrough of mammalian *AGO1* generates a microRNA pathway inhibitor

Anumeha Singh[1], Lekha E Manjunath[1], Pradipta Kundu[2], Sarthak Sahoo[1], Arpan Das[1,†], Harikumar R Suma[1], Paul L Fox[3] & Sandeep M Eswarappa[1,*] [ID]

## Abstract

Translational readthrough generates proteins with extended C-termini, which often possess distinct properties. Here, we have used various reporter assays to demonstrate translational readthrough of *AGO1* mRNA. Analysis of ribosome profiling data and mass spectrometry data provided additional evidence for translational readthrough of *AGO1*. The endogenous readthrough product, Ago1x, could be detected by a specific antibody both *in vitro* and *in vivo*. This readthrough process is directed by a *cis* sequence downstream of the canonical *AGO1* stop codon, which is sufficient to drive readthrough even in a heterologous context. This *cis* sequence has a let-7a miRNA-binding site, and readthrough is promoted by let-7a miRNA. Interestingly, Ago1x can load miRNAs on target mRNAs without causing post-transcriptional gene silencing, due to its inability to interact with GW182. Because of these properties, Ago1x can serve as a competitive inhibitor of miRNA pathway. In support of this, we observed increased global translation in cells overexpressing Ago1x. Overall, our results reveal a negative feedback loop in the miRNA pathway mediated by the translational readthrough product of *AGO1*.

**Keywords** Argonaute; let-7a; miRNA; translational readthrough
**Subject Category** RNA Biology
**The EMBO Journal (2019) 38: e100727**

## Introduction

Translating ribosomes normally terminate at the first in-frame stop codon they encounter on the mRNA. However, in some transcripts, under certain conditions, translating ribosomes continue to translate beyond the stop codon until they encounter another in-frame stop codon. This process, termed as translational readthrough (or stop codon readthrough), generates longer isoforms with unique C-terminal extensions, which can have different functions or different localization (Eswarappa *et al*, 2014; Schueren *et al*, 2014; Hofhuis *et al*, 2016), thus contributing to proteome expansion.

Non-programmed basal translational readthrough that normally occurs at a very low level (~0.1%) is attributed to translation errors (Rajon & Masel, 2011). However, functional translational readthrough is usually programmed by the downstream *cis*-acting RNA elements such that only select transcripts undergo this process. For example, a pseudoknot structure drives translational readthrough in the *gag-pol* region of murine leukemia virus (Houck-Loomis *et al*, 2011). Readthrough can be regulated by *trans*-acting molecules that bind these RNA elements as shown in the case of *VEGFA* where an RNA-binding protein, hnRNPA2/B1, promotes readthrough (Eswarappa *et al*, 2014).

Programmed translational readthrough (PTR) has been observed in viruses, fungi, fruit flies, and mammals. The significance of this process in mammals is largely undetermined (Schueren & Thoms, 2016). Several mammalian genes such as *HBB*, *MPZ*, *VEGFA*, *OPRK1*, *OPRL1*, *AQP4*, *MAPK10*, *LDHB*, *MDH1*, and *VDR* have been experimentally validated as PTR targets (Chittum *et al*, 1998; Yamaguchi *et al*, 2012; Eswarappa *et al*, 2014; Loughran *et al*, 2014, 2018). Several more have been predicted to undergo translational readthrough whose regulation and physiological significance remain to be characterized (Jungreis *et al*, 2011; Dunn *et al*, 2013).

In a genome-wide screen, *AGO1* was identified as a potential readthrough candidate in mammals (Eswarappa *et al*, 2014). *AGO1* encodes Argonaute1 (Ago1) protein, which is a key player in the miRNA-mediated gene silencing. Human genome encodes four Argonaute proteins, Ago1–4. Among them, Ago1, Ago3, and Ago4 are non-catalytic, while Ago2 is catalytic with endonuclease function. Ago proteins interact with miRNAs and siRNAs and load them onto their target mRNAs to silence their expression. In humans, mRNA targets with sequences perfectly complementary to small RNAs are cleaved by Ago2 (Meister, 2013). Partial complementarity between target mRNA and small RNA leads to repression of protein

1 Department of Biochemistry, Indian Institute of Science, Bengaluru, Karnataka, India
2 Department of Microbiology and Cell Biology, Indian Institute of Science, Bengaluru, Karnataka, India
3 Department of Cellular and Molecular Medicine, The Lerner Research Institute, Cleveland Clinic, Cleveland, OH, USA
*Corresponding author. Tel: +91 80 22932881; E-mail: sandeep@iisc.ac.in
†Present address: Department of Molecular, Cellular and Developmental Biology, University of Colorado, Boulder, CO, USA

synthesis and/or mRNA degradation. This process is mediated by GW182 protein, which binds Ago proteins loaded on the mRNA via its N-terminal AGO-binding domain (ABD). GW182 in turn recruits downstream effector proteins such as cytoplasmic deadenylase complexes resulting in mRNA degradation (Jonas & Izaurralde, 2015). Therefore, Ago proteins are essential for miRNA and siRNA-mediated post-transcriptional gene silencing. They are highly conserved proteins and found in most eukaryotes (except *Saccharomyces cerevisiae*). A recent report shows that Argonaute–miRNA complexes can target mRNAs even in the nucleus of stem cells (Sarshad *et al*, 2018). In addition to their role in small RNA-mediated post-transcriptional gene silencing, human Ago proteins have roles in transcription, splicing, and DNA repair (Meister, 2013). A recent unpublished study uploaded on the preprint server bioRxiv shows that translational readthrough product of *AGO1* (Ago1x) is expressed in breast cancer cells and it prevents dsRNA-induced interferon signaling. This nuclear function of Ago1x is not related to miRNA pathway (preprint: Ghosh *et al*, 2019).

The function of Ago proteins is regulated by post-translational modifications such as phosphorylation, ubiquitination, hydroxylation, and poly(ADP-ribose) modification (Qi *et al*, 2008; Rybak *et al*, 2009; Leung *et al*, 2011; Rudel *et al*, 2011). Here, we report the regulation of *AGO1* at translational level. Our results demonstrate PTR of the *AGO1* transcript, which results in an isoform termed as Ago1x. This novel isoform does not cause post-transcriptional gene silencing due to its inability to interact with GW182. Interestingly, this process is positively regulated by let-7a miRNA. Thus, our study uncovers a negative feedback mechanism to dampen the miRNA pathway.

# Results

## AGO1 transcript undergoes programmed translational readthrough

Predicted amino acid sequence encoded in the proximal part of the 3′UTR (untranslated region) of *AGO1* shows remarkable evolutionary conservation within mammals. Also there is a stop codon downstream of and in-frame with the canonical stop codon in the 3′UTR, which is evolutionarily conserved in several mammals. These two stop codons are separated by 99 nucleotides, which potentially encode 33 amino acids in humans (Fig 1A). Based on these features, *AGO1* was identified as a possible candidate for translational readthrough (Eswarappa *et al*, 2014). To confirm the readthrough process, we employed a luciferase-based translational readthrough assay. Partial coding sequence of human *AGO1* (696 nucleotides at the 3′ end) was cloned along with the canonical stop codon and the inter-stop codon region (ISR), upstream of and in-frame with the coding sequence of firefly luciferase (FLuc) (*AGO1*-TGA-ISR-*FLuc*). The downstream in-frame stop codon in the 3′UTR of *AGO1* and the start codon of FLuc were not included in the construct such that FLuc is expressed *only* if there is translational readthrough across the canonical stop codon of *AGO1*. A construct without any stop codon between *AGO1* and FLuc was used to quantify the percentage of readthrough (*AGO1*-GCA-ISR-*FLuc*) (Fig 1B). These constructs were transfected in HEK293 cells, and the translational readthrough was measured as FLuc activity normalized to the activity of co-transfected Renilla luciferase (RLuc). We observed significantly higher relative luciferase activity in the construct with ISR compared to the one without it (*AGO1*-TGA-ΔISR-*FLuc*) indicating programmed translational readthrough of *AGO1*. The readthrough efficiency in this assay was about 20%, and it was comparable to the efficiency of readthrough of *VEGFA*. All three constructs were transcribed to a comparable level in the transfected cells as shown by RT–PCR (Figs 1C and EV1A).

Interestingly, the readthrough process was stop codon-dependent. When the canonical TGA stop codon of *AGO1* was mutated to TAA or TAG, we observed a significant decrease in the readthrough activity (Fig 1D). This observation is consistent with the fact that TGA is the "leakiest" stop codon (Dabrowski *et al*, 2015). We could also demonstrate translational readthrough of *AGO1 in vitro* using rabbit reticulocyte lysate-mediated translation system. Using this assay, we found that the first 57 nucleotides of the ISR were enough to induce readthrough (Fig 1E). The *in vitro* assay also revealed the importance of TGA stop codon during translational readthrough of *AGO1*, in agreement with the transfection experiment (Fig EV1B).

**Figure 1. AGO1 transcript undergoes translational readthrough.**

A   Alignment of amino acid sequences potentially encoded by the proximal 3′UTR of *AGO1* mRNA. Conserved residues are shown in gray background; sequence of the peptide used to raise an antibody against the putative readthrough product (Ago1x) is shown. *, position of in-frame stop codons.

B   Schematic of the construct used in luciferase-based readthrough assays. The last 696 nucleotides of coding sequence of *AGO1* and 99 nucleotides of the ISR were cloned upstream and in-frame with firefly luciferase (FLuc).

C   Demonstration of translational readthrough of *AGO1* using luciferase-based reporter assay. Plasmids containing in-frame *AGO1*-TGA (or GCA)-ISR-*FLuc* were transfected in HEK293 cells, and translational readthrough was detected as FLuc activity normalized to the activity of co-transfected Renilla luciferase (RLuc). *FLuc* mRNA levels determined by RT–PCR are shown below. ***$P$ = 0.0001.

D   Significance of the identity of the canonical stop codon in *AGO1* translational readthrough. AGO1-ISR-FLuc constructs containing TGA or TAA or TAG were transfected in HEK293 cells, and translational readthrough was quantified as described above. *FLuc* mRNA levels determined by RT–PCR are shown below. ***$P$ = 0.0002; **$P$ = 0.0004.

E   The first 57 nucleotides of ISR are sufficient to drive *AGO1* translational readthrough. *AGO1*-TGA-*FLuc* constructs with different lengths of ISR (all in-frame with *AGO1* and *FLuc*) were *in vitro* transcribed and *in vitro* translated using rabbit reticulocyte lysate. FLuc activity reflects readthrough activity. **$P$ = 0.012; *$P$ = 0.017.

F   Demonstration of translational readthrough of *AGO1* using fluorescence-based reporter assay. Plasmids containing in-frame *AGO1*-TGA (or GCA)-ISR-*GFP* were transfected in HEK293 cells, and translational readthrough was detected as fluorescence. Scale bar, 50 μm.

G   The bar graph shows mean fluorescence intensities in cells transfected with the indicated constructs. Fluorescence intensity was measured by flow cytometry. ***$P$ = 0.007.

Data information: Bar graphs (mean ± SE) are representative of at least three independent experiments done in triplicate. Statistical significance was calculated using Student's *t*-test. ISR, inter-stop codon region; ΔISR, construct without ISR.

Source data are available online for this figure.

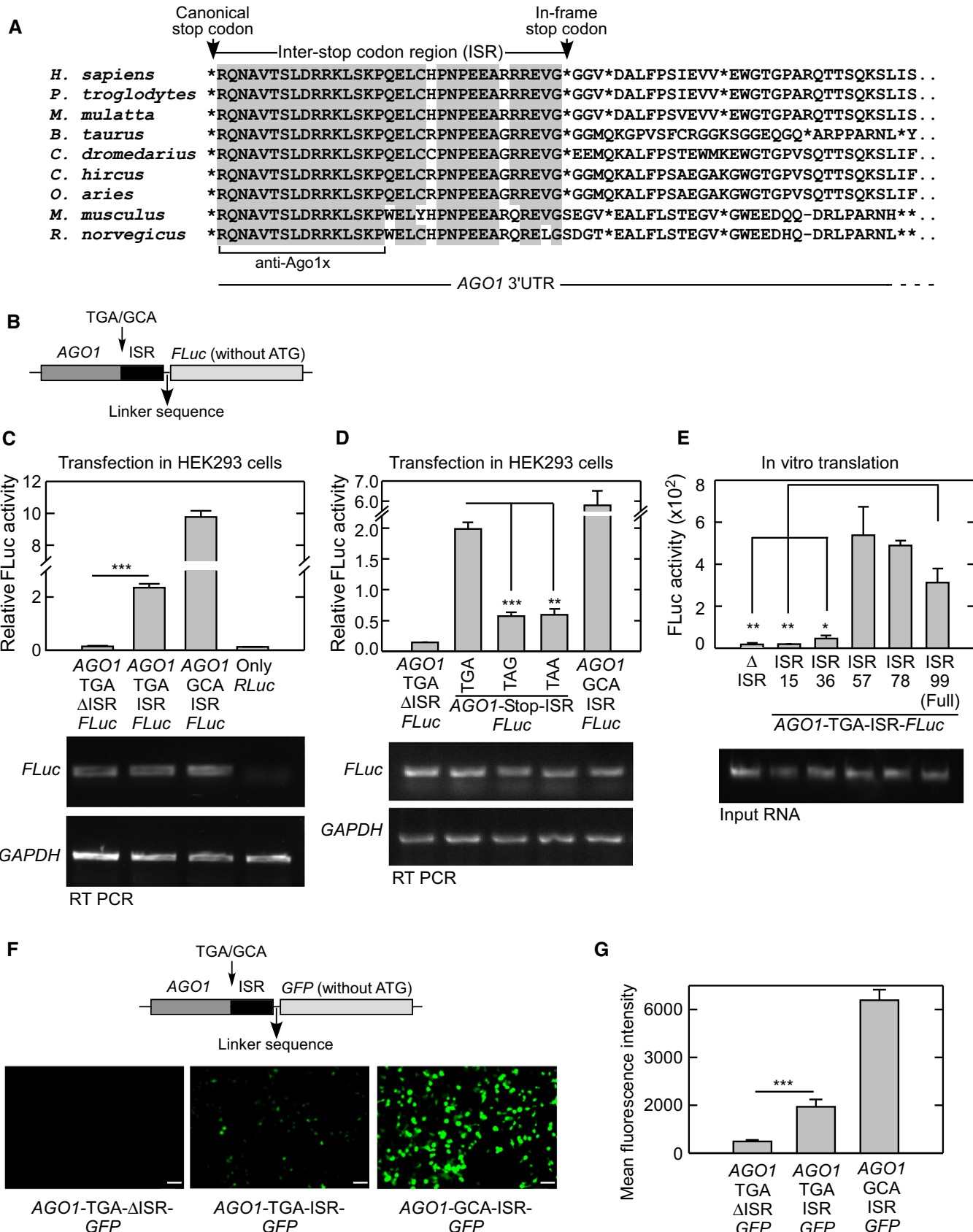

**Figure 1.**

Next, we replaced FLuc in the constructs described above with the Myc-tag coding sequence, such that the translational readthrough generates a Myc-tagged protein. In consistence with the luciferase-based assay, we detected readthrough product of expected molecular weight by Western blot using anti-Myc antibody (Fig EV1C). We raised another antibody (termed anti-Ago1x), which specifically recognizes the peptide RQNAVTSLDRRKLSKP generated only after translational readthrough of *AGO1* (Fig 1A). This antibody also detected the readthrough product. Importantly, these antibodies did not detect the readthrough product in cells transfected with a construct that lacks ISR, showing that it is a programmed process (Fig EV1C). To further confirm this process, we replaced FLuc with green fluorescent protein (GFP) coding sequence without its start codon. GFP is expressed in these constructs only if there is translational readthrough across the canonical stop codon of *AGO1*. We observed fluorescence in HEK293 cells transfected with construct having *AGO1* ISR (*AGO1*-TGA-ISR-*GFP*) demonstrating readthrough. Fluorescence was detected using fluorescence microscopy, and its intensity was quantified using flow cytometry (Fig 1F and G).

### Detection of endogenous translational readthrough product of *AGO1* (Ago1x)

Ago1x is 34 amino acids longer at the C-terminus compared to Ago1 (1 encoded by the canonical stop codon and 33 by the ISR) (Fig 2A). Anti-Ago1x antibody was raised to detect a peptide (RQNAV TSLDRRKLSKP) in this unique region of Ago1x (see above). This antibody could detect a protein of expected size of ~100 kDa, in HEK293 and HeLa cell lysates. Importantly, this band was not found when probed with pre-immune serum or the peptide-pretreated antibody (Fig 2B and C). This antibody could also detect the overexpressed Ago1x protein (Fig EV2A). To further confirm its specificity, we used anti-Ago1x antibody to probe Ago1x in HEK293 cells transfected with *AGO1*-specific siRNAs. In these cells, the intensity of ~100-kDa band detected by anti-Ago1x antibody was reduced, showing that the band indeed represents an isoform of *AGO1* (Fig 2D). To resolve Ago1 and Ago1x on the same gel, HEK293 cell lysate was electrophoresed for prolonged duration (3.5 h) in a 10% SDS-polyacrylamide gel. Western blot of this gel using anti-Ago1 antibody (which targets the N-terminus of Ago1 isoforms) showed both canonical isoform and the readthrough product as distinct bands. Both isoforms were found as doublets indicating post-translational

modification reported previously for Ago proteins (Qi *et al*, 2008; Leung *et al*, 2011; Rudel *et al*, 2011). Quantification of the bands in the Western blot indicated that Ago1x represents about 40% of the total Ago1 protein in HEK293 cells under the conditions tested (Fig 2E). We could also detect Ago1x in the brain, heart, kidney and the skeletal muscle of mouse, showing that the PTR of *AGO1* takes place *in vivo*. Ago1x expression was particularly higher in brain compared to other tissues tested (Fig 2F).

### Evidence of translational readthrough of endogenous *AGO1* in ribosome profiling data and mass spectrometry data

To obtain more evidence for translational readthrough of *AGO1*, we analyzed ribosome profiling data available in NCBI's Sequence Read Archive (SRA). Ribosome profiling reveals mRNA regions that are protected by translating ribosomes. Hence, ribosomal footprints found after the canonical stop codon and within the ISR of *AGO1* suggest translational readthrough. Using a modified version of previously reported criteria (Dunn *et al*, 2013) (see the Materials and Methods section), we found evidence for translational readthrough of *AGO1* transcript in the ribosome profiling data from U2-OS, MCF7, Huh7, HCT116, HeLa, HEK293, and murine splenic B cells. Moreover, we could detect Ago1x by Western blot in these cell lines (Fig 2G, Appendix Fig S1). Sixteen ribosome profiling data files from U2-OS cells showed strong evidence of translational readthrough (a > 20-fold increase in ribosomal density in the ISR compared to 3'UTR; Table EV1). Therefore, we performed 3-nucleotide periodicity for the ribosome profiling data from U2-OS cells (Elkon *et al*, 2015). The distribution of reads in the ISR was non-uniform, and it was similar to coding sequence with a majority of them in the $0^{th}$ frame, which is consistent with translational readthrough of *AGO1* (Fig EV2B).

Since Ago1x expression was observed in mouse tissues (Fig 2F), we next analyzed previously published high-resolution mass spectrometry data obtained from the mouse tissues for the evidence of *AGO1* translational readthrough (Azimifar *et al*, 2014; Deshmukh *et al*, 2015; Sharma *et al*, 2015). The mass spectrometry data were retrieved from the ProteomeXchange Consortium and analyzed using MaxQuant as described in the Materials and Methods section (Cox & Mann, 2008). We observed Ago1x-specific peptides (encoded by the ISR of *AGO1*) in the mass spectrometry data from mouse brain, skeletal muscle, and liver (Table EV2 and Appendix Fig S2).

---

**Figure 2. Detection of endogenous readthrough product of *AGO1* (Ago1x).**

A   Schematic showing the difference between two isoforms of *AGO1*.

B   Detection of endogenous Ago1x by Western blot in HEK293 cells using anti-Ago1x antibody and pre-immune serum.

C   Western blot of lysates from HEK293 and HeLa cells using anti-Ago1x antibody pretreated with Ago1x-specific peptide.

D   Expression of Ago1x in HEK293 cells transfected with three *AGO1*-specific siRNAs (1, 2, and 3). qRT–PCR analysis of *AGO1* transcript expression in these cells is shown below. ***$P < 0.0001$ (Student's *t*-test). Bars: mean $\pm$ SE; $n = 3$.

E   Proportion of Ago1x isoform in HEK293 cells. To separate Ago1x and Ago1 (~3 kDa difference), the cell lysate was electrophoresed for 3.5 h in 10% polyacrylamide gel. The proportion of Ago1x and canonical isoform (Ago1) was calculated by densitometric analysis. Results from three independent experiments are shown as a bar graph (mean $\pm$ SE).

F   Expression of Ago1x in mouse organs.

G   Presence of ribosome footprint in the inter-stop codon region of *AGO1* transcript in U2-OS cells. Ribosome profiling data were taken from NCBI's Sequence Read Archive (SRA accession no. SRR1257257), and ribosome footprints on *AGO1* transcript were analyzed as described in the Materials and Methods section. Zoomed area near the inter-stop codon region is shown in the inset. Positions of the canonical stop codon and the downstream in-frame stop codon are shown. Detection of Ago1x by Western blot in U2-OS cells is shown (right).

Source data are available online for this figure.

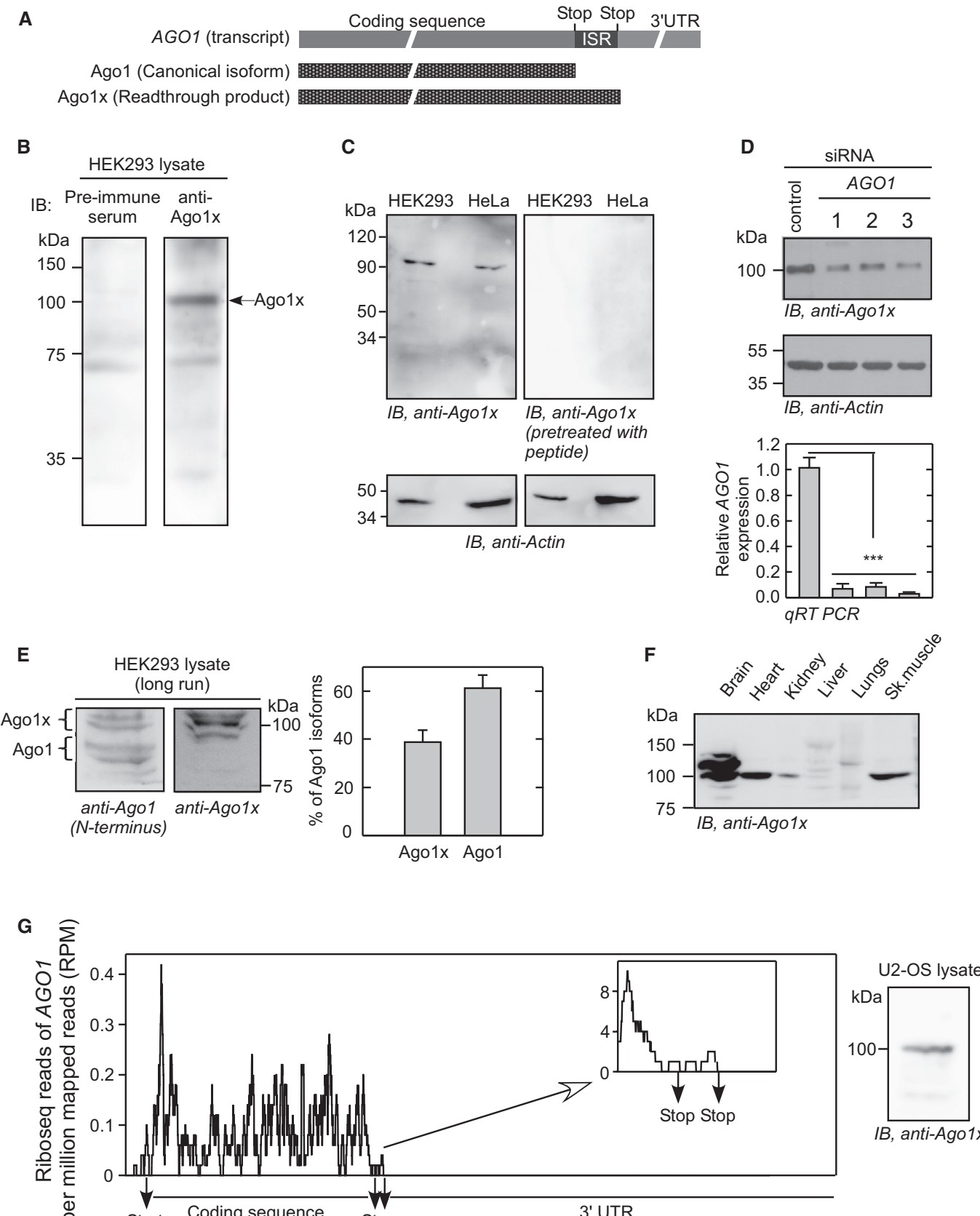

**Figure 2.**

This observation provides a direct evidence for the translational readthrough of endogenous *AGO1*.

### Let-7a miRNA promotes translational readthrough of *AGO1*

There is a putative let-7a miRNA-binding site in the ISR of *AGO1* (Chen *et al*, 2013). This site is present 10 nucleotides downstream of the canonical stop codon (Fig 3A). It is important to note that terminating ribosomes occupy 6 nucleotides after the stop codon (Ingolia *et al*, 2009). These observations suggest a role for let-7a miRNA in the regulation of translational readthrough of *AGO1*. To test this hypothesis, we used let-7a miRNA-specific inhibitor in HeLa cells, which express abundant let-7a miRNA (Shell *et al*, 2007). let-7a miRNA inhibitor is a double-stranded small RNA molecule that binds specifically to let-7a and inhibits its function. Using a reporter containing three let-7a binding sites in the 3′UTR of Renilla luciferase, we confirmed the inhibitory activity of let-7a miRNA inhibitor (Appendix Fig S3A). Interestingly, we observed decreased expression of Ago1x in HeLa cells transfected with let-7a miRNA inhibitor. Surprisingly, there was no change in the level of total Ago1, which was detected using an antibody that targets the N-terminus of Ago1 isoforms (Fig 3B). In consistence with this, overexpression of let-7a miRNA increased the expression of Ago1x without altering the total Ago1 protein level (Fig 3C and Appendix Fig S3B). To further confirm this, we employed luciferase-based translational readthrough assay in HeLa cells (Fig 1B). Cells treated with let-7a miRNA inhibitor showed reduced readthrough activity compared to control cells (Fig 3D). We then mutated the let-7a binding site of the ISR in these constructs. Cells transfected with mutant ISR showed reduced readthrough activity (Fig 3E). Similarly, there was reduced readthrough when the let-7a binding site was moved 18 nucleotides away from the stop codon in the ISR (Fig 3F). To determine whether the *AGO1* ISR that contains a let-7a binding site can drive readthrough in a heterologous context, we replaced *AGO1* coding sequence with *RHOA* coding sequence (582 nucleotides) or Renilla luciferase coding sequence in the constructs used for luciferase-based readthrough assay. Interestingly, *AGO1* ISR alone could drive translational readthrough of *RHOA* and Renilla luciferase across their stop codon (Fig 4A and B). Together, these results show that let-7a miRNA promotes translational readthrough of *AGO1* mRNA without altering its canonical translation.

The fact that rabbit reticulocyte lysate contains abundant let-7a miRNA explains efficient translational readthrough of *AGO1* observed in the lysate (Figs 1E and EV1B) (Ricci *et al*, 2011).

We adopted the BoxB–N-peptide tethering system to further confirm the miRNA-regulated translational readthrough of *AGO1*. BoxB is a stem-loop RNA element that interacts with a 22-amino acid peptide called N-peptide. This interaction is important for transcription anti-termination in bacteriophage λ. The BoxB–N-peptide tethering system has been used to investigate miRNA function (Pillai *et al*, 2004). We cloned two BoxB elements downstream of the *AGO1* coding sequence and its stop codon followed by luciferase coding sequence. There was no *AGO1* ISR in the construct. BoxB element will bind N-peptide-tagged proteins. When this construct was co-transfected with N-peptide-tagged Ago1 or Ago1x, there was a significant induction of readthrough as indicated by increased luciferase activity (Fig 4C). This observation supports our conclusion that let-7a miRNA can promote readthrough of *AGO1* mRNA.

### Ago1x can load miRNAs on target mRNAs

Having established the PTR of *AGO1*, we investigated the properties of the novel isoform, Ago1x, generated by this process. Since translational readthrough products can have different intracellular localization compared to canonical isoforms, we first examined the intracellular distribution of these two isoforms. Similar to Ago1, Ago1x showed punctate distribution in HeLa cells without any obvious difference between them (Appendix Fig S4). Next, we analyzed the interaction of Ago1x with miRNAs. HeLa cells were transfected with FLAG-HA-tagged Ago1 or FLAG-HA-tagged Ago1x. For all Ago1x overexpression studies, we changed the stop codon TGA to TCA (serine codon) to achieve overexpression comparable to that of Ago1. TCA was chosen because serine is one of the amino acids encoded by TGA stop codon during translational readthrough (Hatfield, 1972; Chittum *et al*, 1998; Eswarappa *et al*, 2014). The FLAG-tagged protein was immunoprecipitated using anti-FLAG beads. Co-immunoprecipitated miRNAs were analyzed by deep sequencing of small RNAs. About 93% (395/424) of Ago1-interacting miRNAs and 99% (395/397) of Ago1x-interacting miRNAs were the same (Appendix Tables S1–S3). Linear regression analysis of the miRNA read values (reads per million) indicated that interaction of Ago1x with miRNAs is comparable to that of Ago1 (Fig 5A). Quantitative

---

**Figure 3.   Translational readthrough of *AGO1* is regulated by let-7a miRNA.**

A   Sequence alignment showing the binding site of let-7a miRNA in the inter-stop codon region of *AGO1*.

B   Effect of let-7a inhibitor on translational readthrough of *AGO1*. let-7a inhibitor or control inhibitor was transfected in HeLa cells, and the expression of Ago1x was determined by Western blot after 48 h. Densitometric analysis of three independent experiments is shown as a bar graph (mean ± SE). *$P$ = 0.01.

C   Effect of let-7a overexpression on translational readthrough of *AGO1*. HeLa cells were stably transfected with the pri-let-7a overexpression construct. Ago1x expression was determined by Western blot. Densitometric analysis of three independent experiments is shown as a bar graph (mean ± SE). *$P$ = 0.007.

D   let-7a inhibitor reduces the *AGO1* readthrough efficiency. HeLa cells were transfected with let-7a inhibitor or control inhibitor along with the *AGO1*-TGA (or GCA)-ISR-*FLuc* construct. Readthrough was quantified by measuring FLuc activity, which was normalized to the activity of co-transfected Renilla luciferase. *FLuc* mRNA expression in HeLa cells treated with miRNA inhibitors is shown (Right). *$P$ = 0.027.

E   *AGO1* ISR with mutated let-7a binding site shows reduced translational readthrough activity. Assay was done as described above in HeLa cells. Mutated sequence is shown (top). ***$P$ = 0.008.

F   *AGO1* ISR with displaced let-7a binding site (*AGO1*-TGA-disISR-*FLuc*) shows reduced translational readthrough activity. let-7a binding site was moved 18 nucleotides downstream of the stop codon. Readthrough assay was done as described above in HeLa cells. ***$P$ < 0.0001.

Data information: Bar graphs (mean ± SE) are representative of at least three independent experiments. Statistical significance was calculated using Student's *t*-test (B and C, paired *t*-test; others, unpaired *t*-test).
Source data are available online for this figure.

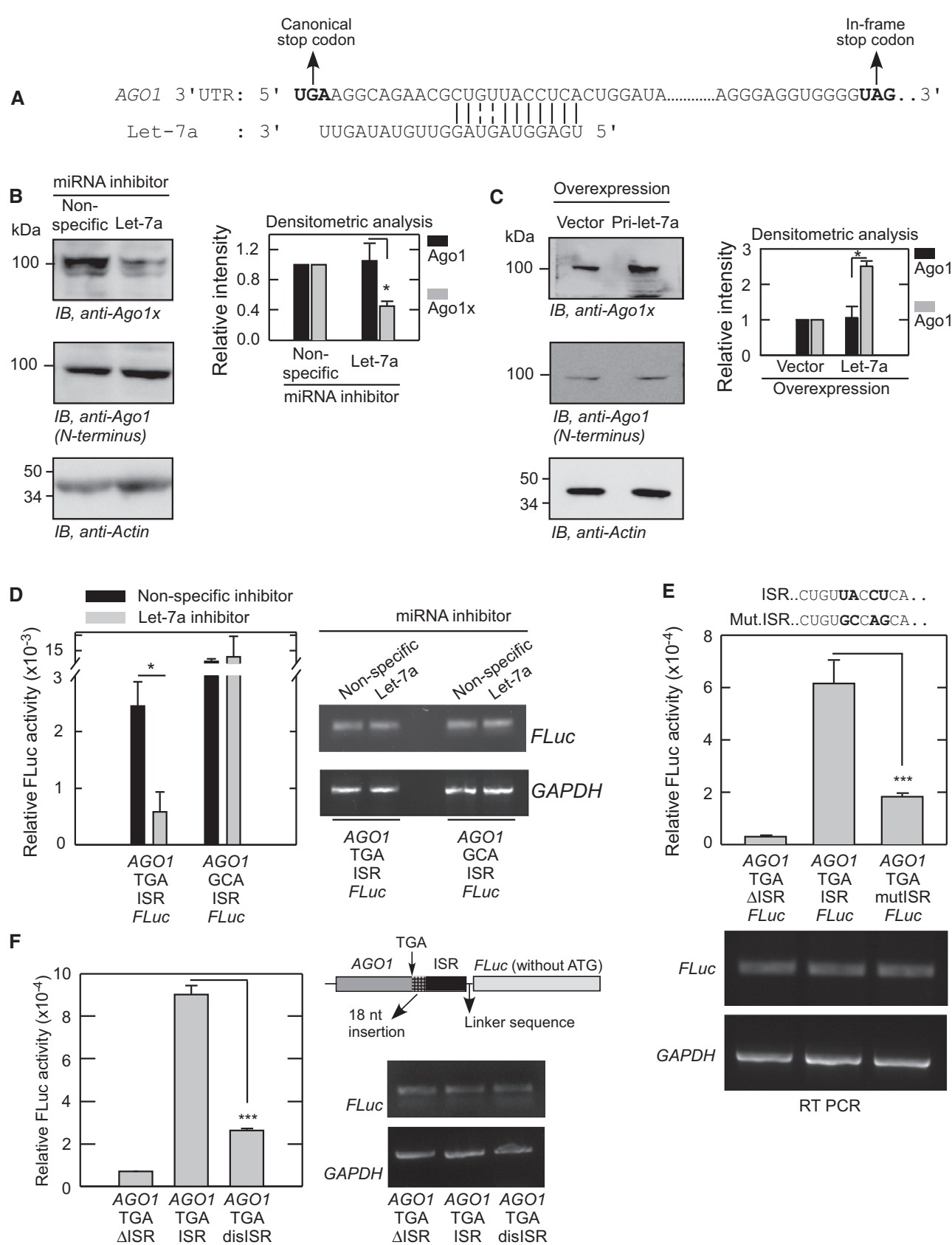

**Figure 3.**

real-time PCR also revealed a comparable interaction of Ago1 and Ago1x with seven miRNAs (Fig 5B). Immunoprecipitation followed by RT–PCR of co-immunoprecipitated RNA samples for *MYC* and *PTEN* transcripts, two known targets of miRNA-mediated regulation (Sampson *et al*, 2007; Mouw *et al*, 2014), revealed that Ago1x can also interact with these mRNAs (Fig 5C). Like Ago1, Ago1x was also found to be interacting with Dicer protein, which is a part of RISC (RNA-induced silencing complex)-loading complex (RLC). RLC recruits miRNAs to Ago proteins, and this step is critical for the miRNA function (Fig 5D). Together, these results show that Ago1x is capable of loading miRNAs onto target mRNAs. These interactions were observed even in cells where expression of endogenous Ago1 was knocked down using a specific shRNA that targets its 3′UTR. This shows that the ability of Ago1x to load miRNAs onto target mRNAs is independent of Ago1 (Fig EV3A–D).

## Ago1x does not repress translation of target mRNAs

Ago proteins and their associated miRNA are part of RISC, which also contains other components critical for the silencing of gene expression (Jonas & Izaurralde, 2015). One such component is GW182 (or TNRC6A) that mediates translational repression and degradation of target mRNAs by recruiting several effector proteins (Pfaff & Meister, 2013). Interestingly, GW182 protein was co-immunoprecipitated along with Ago1, but not with Ago1x (Fig 6A). This lack of interaction between Ago1x and GW182 was observed even in cells where endogenous Ago1 was knocked down (Fig EV4A). Further, Ago1, but not Ago1x, was co-immunoprecipitated with endogenous GW182 (Fig EV4B). In support of this, confocal immunofluorescence images showed significant co-localization of GW182 with Ago1 puncta (87% out of > 50 cells), but its co-localization with Ago1x puncta was greatly reduced (only 5% out of > 50 cells) (Fig 6B). In order to confirm the differential interaction of GW182 with both the Ago1 isoforms, we performed proximity ligation assay (PLA), which detects protein–protein interactions with high sensitivity and specificity (Gustafsdottir *et al*, 2005; Soderberg *et al*, 2006). This assay also showed that GW182 interacts with Ago1, but not with Ago1x, as indicated by the highly reduced number of dots generated after proximity ligation in Ago1x-transfected cells compared to Ago1-transfected cells (Fig 6C). Together, these results show that unlike Ago1, Ago1x does not interact with GW182.

One of the functions of GW182 is to mediate translational repression of target mRNAs. In fact, GW182 proteins are essential for

miRNA-mediated gene silencing. Lack of GW182 proteins drastically inhibits silencing in *C. elegans*, *D. melanogaster*, and mammalian cells (Rehwinkel *et al*, 2005; Eulalio *et al*, 2008; Jonas & Izaurralde, 2015). This implies that Ago1x, unlike Ago1, cannot repress translation of target mRNAs. To test this hypothesis, we employed the BoxB–N-peptide tethering system described above. Ago proteins tagged with N-peptide can bind transcripts containing BoxB elements and bring down their translation (Fig 7A) (Pillai *et al*, 2004, 2005). We used a RLuc construct with five BoxB elements in its 3′UTR. This RLuc–BoxB construct was transfected in HeLa cells along with N-peptide-tagged or untagged Ago1 or Ago1x or Ago2. After achieving comparable expression of these Ago proteins, RLuc translation was measured as RLuc activity relative to the activity of co-transfected FLuc. As expected, we observed a significant decrease in RLuc activity in cells transfected with N-peptide-tagged Ago1 and Ago2. However, N-peptide-tagged Ago1x did not show any decrease in RLuc activity; rather, there was an increase (Fig 7B). To further confirm these observations, we compared the level of HA-tagged RLuc protein by Western blot. Similar to RLuc activity, RLuc protein expression was reduced in cells transfected with N-peptide-tagged Ago1, but not in cells transfected with N-peptide-tagged Ago1x. RLuc transcript level was comparable in all the conditions studied (Fig 7C). We then transferred the ISR of *AGO1* to downstream of *AGO2* such that a chimeric protein Ago2ISR$^{Ago1}$ is generated. Unlike the N-peptide-tagged Ago2, N-peptide-tagged Ago2ISR$^{Ago1}$ did not exhibit translational repression activity (Fig 7D). Furthermore, unlike Ago2 and similar to Ago1x, Ago2ISR$^{Ago1}$ did not show interaction with GW182 in proximity ligation assay (Fig EV4C). These results show that Ago1x lacks the ability to repress translation because of the extended C-terminal region encoded by the ISR.

## Ago1x is a global miRNA pathway inhibitor

Since Ago1x binds miRNAs, Dicer, and the target mRNAs without repressing the gene expression, it can potentially serve as a competitive inhibitor of miRNA pathway. To investigate this possibility, we studied the global translation in HeLa cells overexpressing FLAG-HA-tagged Ago1 and FLAG-HA-tagged Ago1x using the ribopuromycylation (RPM) method (Schmidt *et al*, 2009; David *et al*, 2012). In this method, incorporation of puromycin by translating machinery is quantified using anti-puromycin antibody. We observed significantly elevated global translation in cells overexpressing FLAG-HA-Ago1x (Fig 8A). Though there was a slight increase in global

---

**Figure 4.  Translational readthrough in a heterologous context.**

A   *AGO1* ISR can drive translational readthrough in a heterologous context of *RHOA*. *AGO1* coding sequence in the *AGO1-TGA-ISR-FLuc* construct was replaced by *RHOA* coding sequence (582 nucleotides), and the readthrough assay was done as described above in HeLa cells. 3′UTR$^{Rho}$, 159 nucleotides in the proximal 3′UTR of *RHOA* (NM_001664); ISR$^{Ago1}$, inter-stop codon region of *AGO1*. \*\*\*$P < 0.0001$.

B   *AGO1* ISR can drive translational readthrough in a heterologous context of Renilla luciferase (*RLuc*). *AGO1* coding sequence in the *AGO1-TGA-ISR-FLuc* construct was replaced by *RLuc* coding sequence, and the readthrough assay was done as described above in HeLa cells (left) and *in vitro* using the Rabbit Reticulocyte Lysate System (right). \*\*$P = 0.006$; \*\*\*$P = 0.0001$.

C   Tethering of N-peptide-tagged Ago1 or Ago1x can induce readthrough of *AGO1*. Two BoxB elements were cloned downstream of the *AGO1* and upstream of the in-frame *FLuc* as shown in the schematic. This construct was transfected in HeLa cells expressing N-peptide-tagged or FLAG-tagged Ago proteins as indicated. Readthrough was measured as described above. \*$P = 0.035$ (with Welch's correction); \*\*$P = 0.016$.

Data information: Bar graphs (mean ± SE) are representative of at least three independent experiments done in triplicate. Statistical significance was calculated using Student's *t*-test.

Source data are available online for this figure.

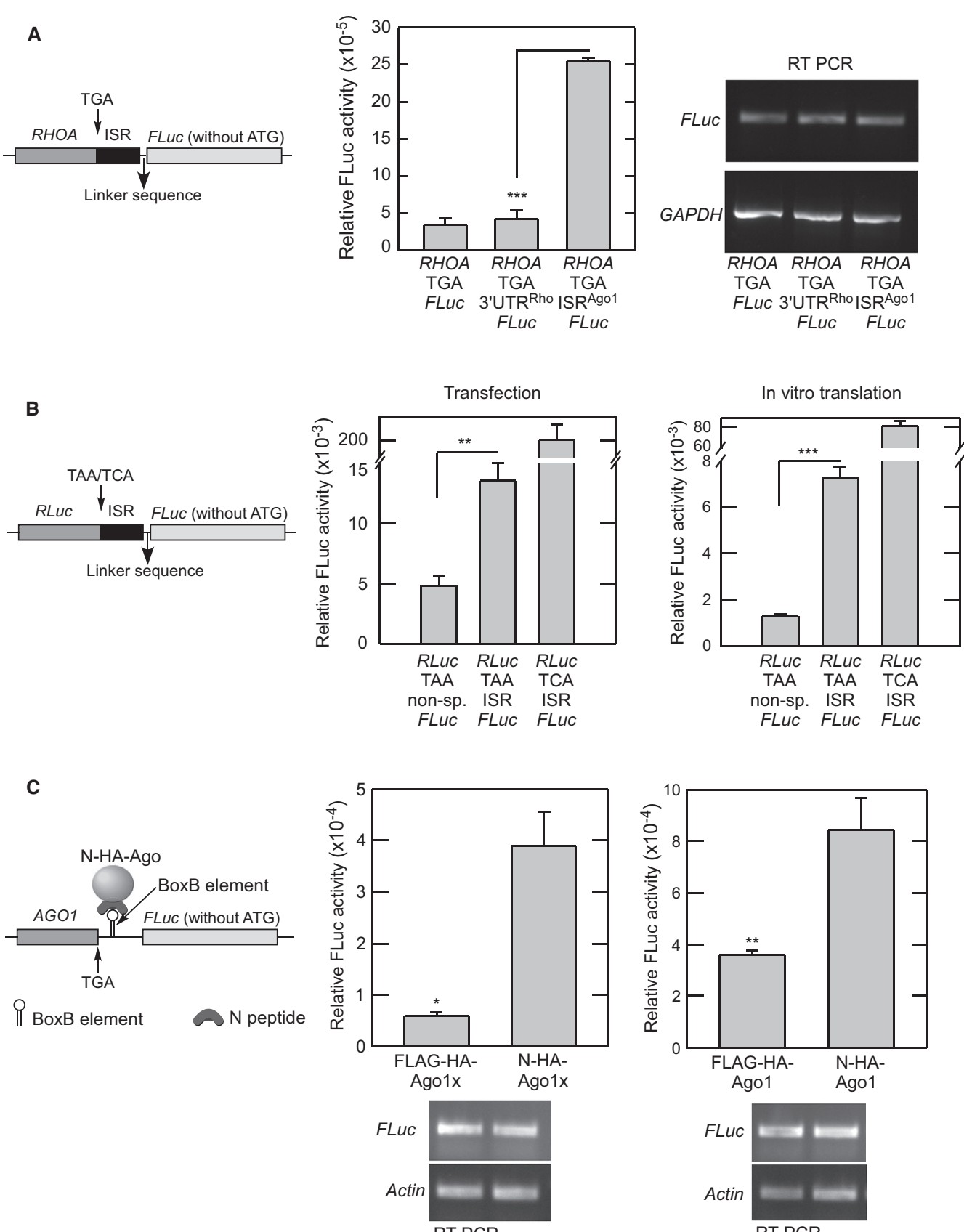

**Figure 4.**

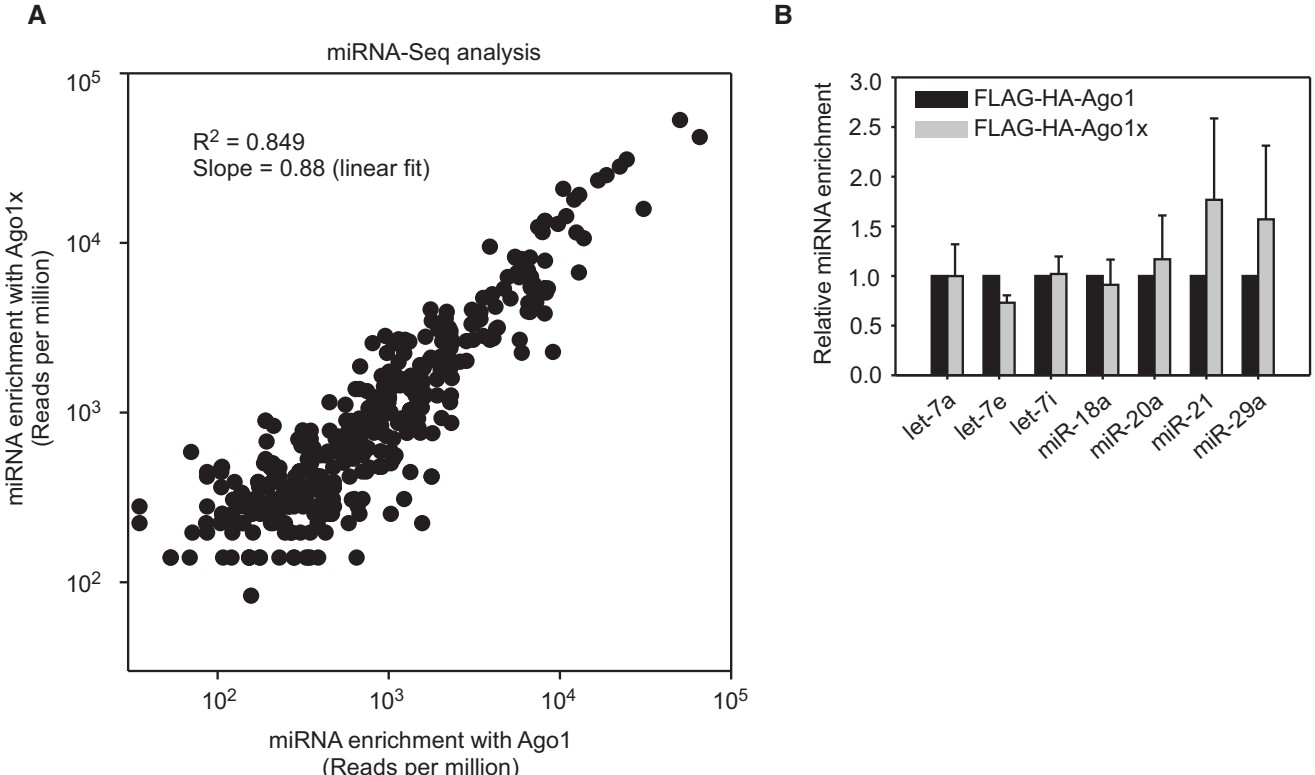

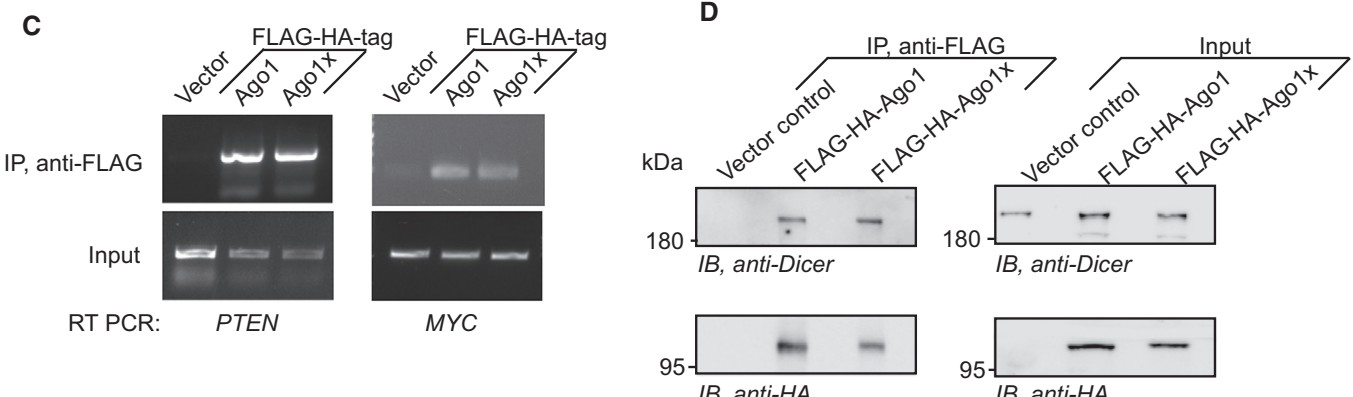

**Figure 5. Ago1x can load miRNAs onto target mRNA.**

A   Linear regression analysis of miRNAs co-immunoprecipitated with Ago1 and those co-immunoprecipitated with Ago1x in HeLa cells. Reads-per-million (RPM) values of 395 miRNAs that were common to Ago1 and Ago1x are shown.

B   qRT–PCR results showing the enrichment of seven miRNAs in Ago1x immunoprecipitate relative to that in Ago1 immunoprecipitate in HeLa cells. Bars show the mean of three experiments ± SE.

C   RT–PCR results showing *PTEN* and *MYC* mRNAs co-immunoprecipitated with Ago1 or Ago1x in HeLa cells.

D   Result of co-immunoprecipitation experiments demonstrating the interaction of Dicer protein with Ago1 and Ago1x in HeLa cells.

Source data are available online for this figure.

translation of cells overexpressing FLAG-HA-Ago1, the difference was not statistically significant across biological replicates compared to vector-transfected cells. Expectedly, RPM performed in GW182 knockdown cells showed elevated translation, thereby validating this assay (Fig EV5). Global translation was also quantified by pulse-labeling cells with [$^{35}$S]-methionine. Cells overexpressing Ago1x showed more [$^{35}$S]-methionine incorporation compared to control cells and those overexpressing Ago1 (Fig 8B). Together, these results show that elevated cellular level of Ago1x causes increased global translation.

**A**

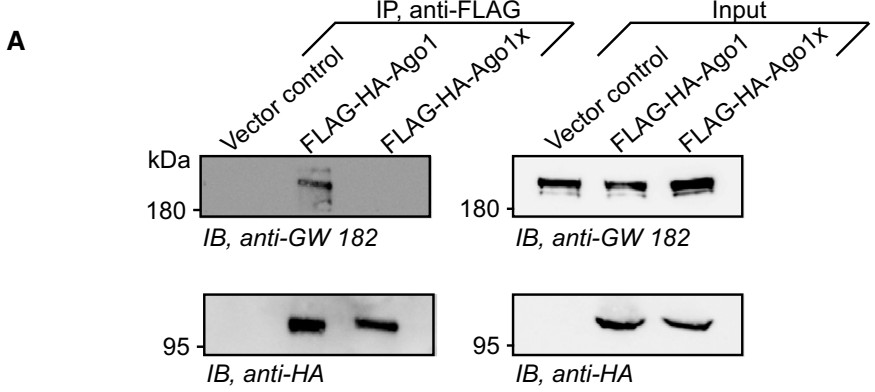

**B**

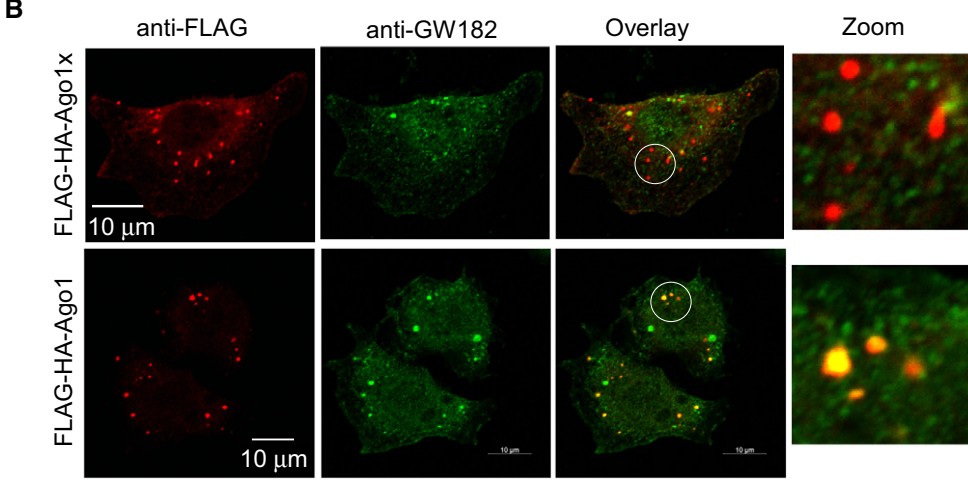

**C**

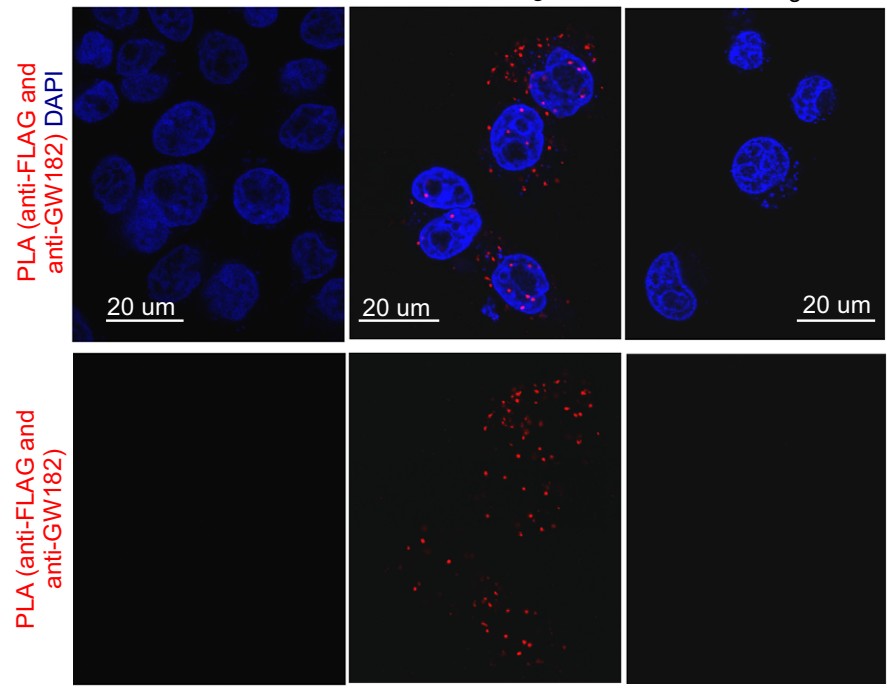

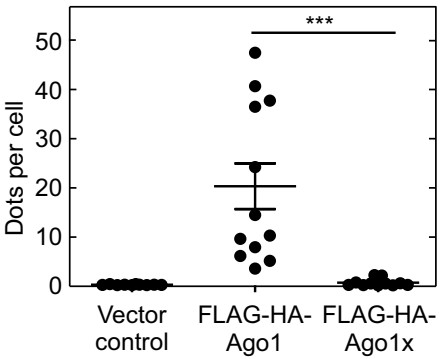

**Figure 6.**

**Figure 6.  Ago1x does not interact with GW182.**

A  Result of co-immunoprecipitation experiment in HeLa cells shows that GW182 is co-immunoprecipitated with Ago1, but not with Ago1x.
B  Confocal immunofluorescence images of HeLa cells show co-localization of GW182 with Ago1, but not with Ago1x. White circles indicate zoomed areas.
C  Confocal microscopy images of proximity ligation assay (PLA) in HeLa cells showing interaction of GW182 with Ago1, but not with Ago1x. Rabbit anti-GW182 and mouse anti-FLAG antibodies were used to achieve proximity ligation. Interaction is indicated by red dots that result from proximity ligation. Quantification of dots per cell is shown (right). Each black circle represents a microscopic field. Means $\pm$ SE are shown as horizontal lines ($n$ = 10 fields for vector control; $n$ = 12 fields for Ago1; $n$ = 11 fields for Ago1x). At least 100 cells were counted for dots in each category. ***$P$ < 0.0001 (Mann–Whitney test).
   All results are representative of three independent experiments.

Source data are available online for this figure.

GW182 is a component of P bodies (processing bodies), which are involved in mRNA degradation and translational repression. Remarkably, the number of P bodies as indicated by GW182-containing puncta in Ago1x-overexpressing cells (2.9 $\pm$ 0.38 per cell) was significantly less ($P$ = 0.004, Mann–Whitney test) compared to that in Ago1-overexpressing cells (4.5 $\pm$ 0.37 per cell). There was no difference in the GW182 expression in these cells (Fig 6A and B). This observation also indicates elevated translation in Ago1x-overexpressing cells.

## Discussion

We provide several lines of evidence for the PTR of mammalian *AGO1* using multiple methods. Luciferase- and fluorescence-based assays not only demonstrated translational readthrough of *AGO1*, but also showed the significance of the identity of the canonical stop codon and the inter-stop codon region in this process. The efficiency of translational readthrough in these experiments varied from 10 to 30% depending on the assay and cells. Western blot using a specific antibody detected endogenous readthrough product (Ago1x) in multiple cell lines and mouse tissues. Analysis of the previously published ribosome profiling data and the mass spectrometry data strengthened the evidence for *AGO1* translational readthrough.

A key finding of this study is that the let-7a miRNA positively regulates translational readthrough of *AGO1*. We demonstrate this using let-7a inhibitor and let-7a overexpression system in both endogenous (Ago1x) and exogenous (luciferase-based assay) systems. We observe robust *AGO1* readthrough in rabbit reticulocyte lysate-mediated *in vitro* translation. This is in agreement with the fact that rabbit reticulocyte lysate (both nuclease-treated and untreated) contains let-7a miRNA (Ricci *et al*, 2011). Similarly, the level of let-7a is relatively high in the brain tissue, which shows

robust Ago1x expression (Ludwig *et al*, 2016) (TissueAtlas). The primary role of miRNAs is to silence the gene expression by translational repression and/or mRNA degradation. miRNAs are also reported to regulate transcription in the nucleus (Liu *et al*, 2018). Interestingly, miR-1224 has been shown to induce ribosomal frameshifting in human *CCR5* mRNA, which encodes a co-receptor for HIV-1 (Belew *et al*, 2014). Our results reveal a new function for miRNA, i.e., regulation of translational readthrough of *AGO1* mRNA. Surprisingly, let-7a inhibition or overexpression did not change the total Ago1 protein level despite a binding site in *AGO1* transcript. This can be explained by the fact that miRNAs that target the coding sequence of mammalian mRNAs with partial complementarity tend to be less efficient due to translating ribosomes than those targeting 3′UTR (Gu *et al*, 2009). Because of translational readthrough, the ISR of *AGO1* is a part of its coding sequence. This could be the reason why let-7a binding there does not affect canonical translation and does not induce mRNA degradation. It is likely that there are other unknown factors, which regulate translational readthrough of *AGO1* in a tissue-specific and cell type-specific manner.

The molecular mechanism of the induction of readthrough by let-7a miRNA remains to be investigated further. The binding site of let-7a miRNA on the ISR of *AGO1* is located ten nucleotides downstream of the canonical stop codon TGA. Incidentally, the binding site of hnRNPA2/B1, a positive regulator of *VEGFA* PTR, on the ISR of *VEGFA* is also situated ten nucleotides downstream of the canonical stop codon (Eswarappa *et al*, 2014). In the case of *gag-pol* region of murine leukemia virus (MLV), the pseudoknot structure that drives translational readthrough is located five nucleotides downstream of the stop codon (Houck-Loomis *et al*, 2011). These observations suggest that ribosomal roadblocks near the stop codon can cause or promote translational readthrough. In fact, translation readthrough and frameshift processes are associated with ribosome pausing (Somogyi *et al*, 1993; Seidman *et al*, 2011). Since

**Figure 7.  Ago1x does not inhibit translation of target mRNAs.**

A  Schematic showing the principle of N-peptide and BoxB element-based assay to study translational repression ability of Ago proteins. N-peptide-tagged Ago1 and Ago2 proteins bind the BoxB elements in the 3′UTR of Renilla luciferase (RLuc) and bring down their translation.
B  Activity of RLuc relative to co-transfected FLuc in HeLa cells expressing N-HA-tagged or untagged Ago proteins as indicated. Bar graph (mean $\pm$ SE) is representative of four independent experiments done in triplicate. Expression of HA-tagged Ago proteins was confirmed by Western blot (bottom). *$P$ = 0.018; #$P$ = 0.041; $P$ = 0.013. Numbers on the bars represent % change in relative luciferase activity.
C  Expression of RLuc protein in HeLa cells expressing N-HA-tagged or untagged Ago proteins as indicated. Densitometric analysis of three independent experiments is shown as a bar graph (mean $\pm$ SE). RT–PCR analysis for RLuc mRNA level is shown (bottom). **$P$ = 0.001.
D  Activity of RLuc relative to co-transfected FLuc in HeLa cells expressing N-HA-tagged Ago2 (N-HA-Ago2) or N-HA-tagged chimeric Ago2 appended with the ISR of Ago1x (N-HA-Ago2-ISR$^{Ago1}$) as indicated. Bar graph (mean $\pm$ SE) is representative of three independent experiments done in triplicate. Expression of HA-tagged Ago proteins was confirmed by Western blot (bottom). ***$P$ < 0.0001; **$P$ = 0.0004.

Data information: Statistical significance in all bar graphs was calculated using Student's *t*-test.
Source data are available online for this figure.

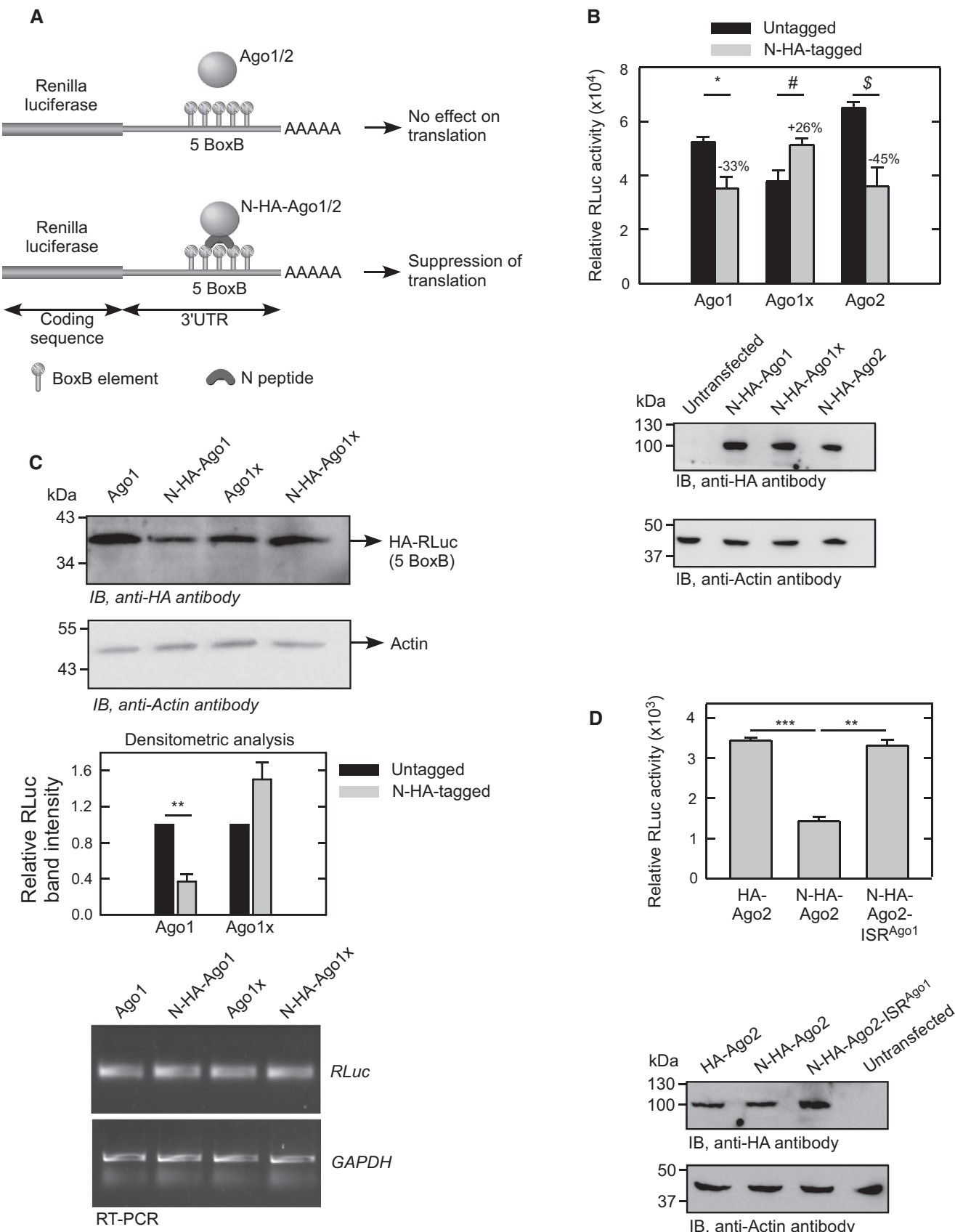

**Figure 7.**

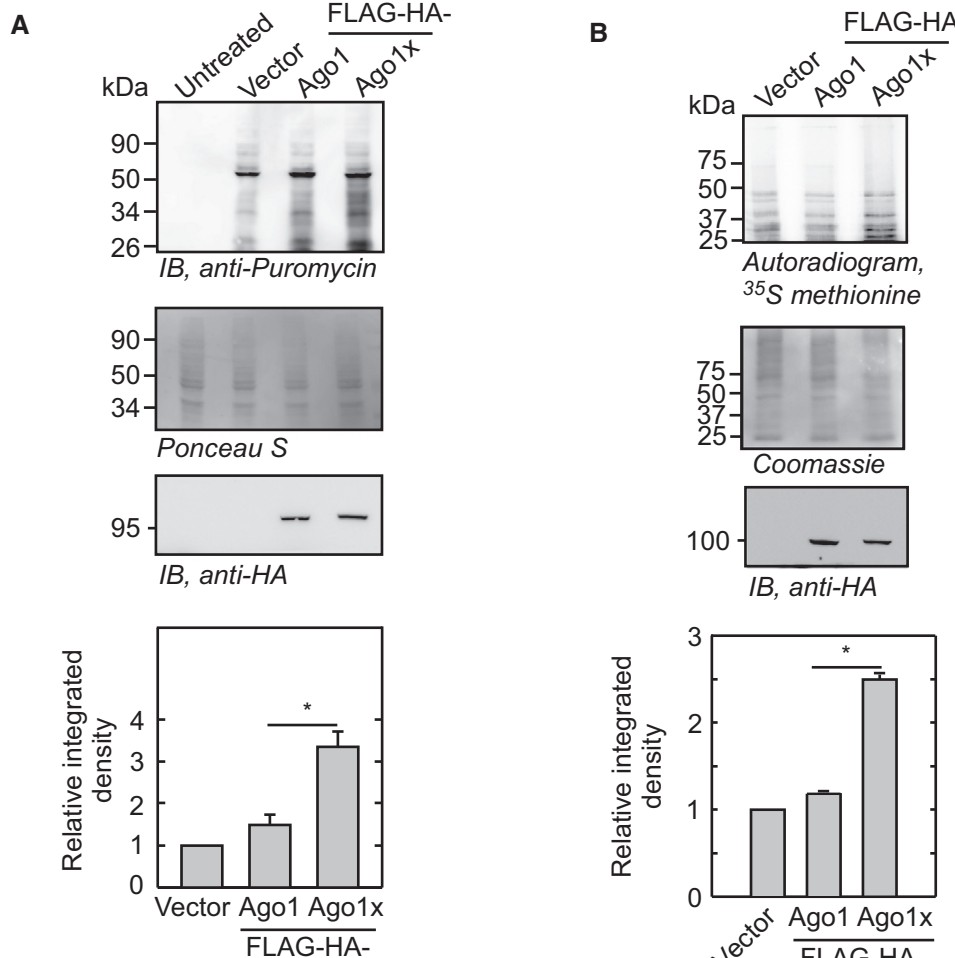

**Figure 8. Overexpression of Ago1x increases global translation.**

A   Ribopuromycylation assay showing global translation profile of HeLa cells overexpressing FLAG-HA-Ago1 or FLAG-HA-Ago1x. Densitometric analysis of four
    independent experiments is shown as a bar graph (mean ± SE). Puromycin densities were normalized to Ponceau S stain densities. *$P < 0.029$ (Mann–Whitney test).
B   [$^{35}$S]-methionine autoradiogram showing global translation profile of HeLa cells overexpressing FLAG-HA-Ago1 or FLAG-HA-Ago1x. Densitometric analysis of three
    independent experiments is shown as a bar graph (mean ± SE). Autoradiogram densities were normalized to Coomassie stain densities. *$P < 0.029$ (Mann–Whitney test).

Source data are available online for this figure.

terminating ribosomes at the stop codon occupy 6 nucleotides after the stop codon, it is possible that these roadblocks (let-7a miRNA in the case of *AGO1*) interact with terminating ribosomes to induce a programmed error resulting in translational readthrough. Structural studies of terminating ribosomes near such roadblocks are required to test this hypothesis.

Ago1x lacks post-transcriptional gene silencing ability because it does not interact with GW182, a protein essential for this effect. However, Ago1x is able to interact with Dicer and load miRNAs on their target mRNAs. Because of these properties, Ago1x acts as a competitive inhibitor of miRNA pathway as shown by increased global translation in cells overexpressing Ago1x (Fig 9). Interestingly, the 34-amino acid C-terminal extension of Ago1x is rich in charged amino acids and is intrinsically disordered as predicted by IUPred web server (Appendix Fig S5) (Dosztanyi *et al*, 2005). This disordered extension may impede the interaction of GW182 with the PIWI domain of Ago1x. This observation is very similar to VEGF-Ax,

readthrough product of *VEGFA*; its C-terminal extension generated after readthrough is also intrinsically disordered and it fails to bind neuropilin, an important co-receptor for the pro-angiogenic function of VEGF-A isoforms (Eswarappa *et al*, 2014).

There are several mechanisms to halt, or even reverse, the miRNA-mediated gene silencing. This reversibility of gene silencing has been observed during stress (Bhattacharyya *et al*, 2006). Post-translational modifications of the Ago proteins are involved in this process. For example, poly(ADP-ribose) modification of Ago2 during stress inhibits the miRNA-guided translational repression and mRNA cleavage (Leung *et al*, 2011). Both Ago1 and Ago2 can undergo phosphorylation, which affects small RNA binding (Rudel *et al*, 2011). In addition, Ago proteins can undergo ubiquitination and hydroxylation, which affects their stability (Qi *et al*, 2008; Meister, 2013). Our study reveals another layer of regulation at the level of translation of *AGO1*. Since let-7a positively regulates translational readthrough of *AGO1* to generate a miRNA pathway inhibitor,

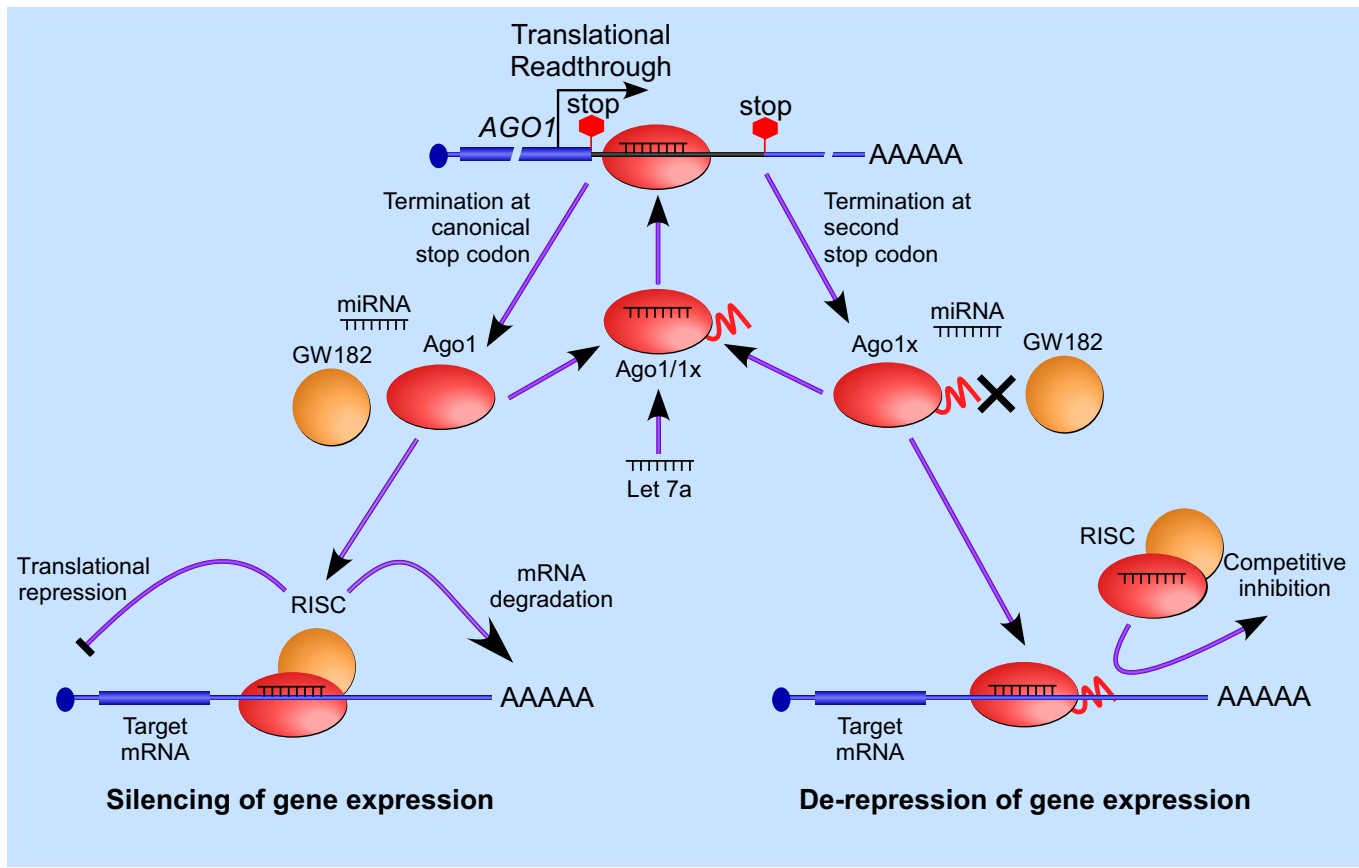

**Figure 9. Schematic showing Ago1x-mediated negative feedback regulation of miRNA pathway.**

Translation termination at the first stop codon of *AGO1* generates canonical Ago1 isoform. Ago1 can load miRNAs on target mRNAs and repress their expression. Translation readthrough at the first stop codon and termination at the downstream in-frame stop codon generates Ago1x. let-7a miRNA promotes this process. Ago1x can load miRNAs on target mRNAs, but it cannot bring down their expression because of its inability to recruit GW182. Thus, it can act as a competitive inhibitor and de-repress gene expression.

Ago1x, it constitutes a negative feedback loop (Fig 9). Interestingly, let-7a constitutes another feedback loop by inhibiting the expression of Dicer (Tokumaru *et al*, 2008). Since Ago1x interacts with 397 miRNAs without mediating gene silencing, it can be considered as a global endogenous miRNA sponge.

# Materials and Methods

## Cell culture

HEK293 and HeLa cells were cultured in Dulbecco's modified Eagle's medium containing 10% fetal bovine serum (Thermo Fisher Scientific), 100 units/ml penicillin, and 100 μg/ml streptomycin (Sigma) at 37°C in a humidified atmosphere containing 5% $CO_2$.

## Amino acid sequence alignment

Nucleotide sequence of the 3′UTR of *AGO1* mRNA from multiple mammalian species was retrieved from the NCBI database. The nucleotide sequence was translated *in silico* to the amino acid sequence using the ExPASy Translate tool. The sequences were then aligned using Clustal Omega.

## Construction of plasmids

The luciferase reporter constructs used in the translation readthrough assays were generated in the pcDNA3.1 backbone. Partial coding sequence of the human *AGO1* (696 nucleotides at the 3′ end) was cloned along with the canonical stop codon and the inter-stop codon region (ISR), upstream of and in-frame with the coding sequence of the firefly luciferase (FLuc). The downstream in-frame stop codon in the 3′UTR of *AGO1* and the start codon of FLuc were not included in the construct such that the FLuc is expressed only if there is translational readthrough across the canonical stop codon of *AGO1*. A linker sequence (GGCGGCTCCGGCGGCTCCCTCGTGC TCGGG) was included upstream of the FLuc sequence to avoid any potential interference between the Ago1 and the FLuc in the fusion protein. PCR-based site-directed mutagenesis was used to generate mutations in the stop codon and the ISR. For displacing the let-7a binding site downstream of the canonical stop codon, ISR of *AGO1* was cloned with an insertion of extra 18 nucleotides (taken from the first twelve nucleotides of ISR).

The open reading frame (ORF) sequence of *AGO1* was PCR-amplified from plasmid containing the full-length cDNA of human *AGO1* (Dharmacon, accession: BC063275, clone ID: 30344513). Plasmids expressing the full-length Ago1 and Ago1x were generated by inserting the *AGO1* and the *AGO1X* coding regions in the pcDNA3.1 backbone between the *Bam*HI and *Xho*I sites. The N-HA-Ago2 and RL-5BoxB constructs have been described in a previous study (Pillai *et al*, 2004). For generating N-HA-Ago1 and N-HA-Ago1x plasmids, annealed DNA oligonucleotides encoding 22-amino acid-long RNA binding domain of the lambda bacteriophage antiterminator protein N (N-peptide) and the influenza hemagglutinin (HA) epitope were inserted in the *Kpn*I and *Bam*HI sites. Plasmids expressing FLAG-HA-Ago1x and FLAG-HA-RL-5BoxB were constructed by replacing the *AGO2* sequence from the FLAG-HA backbone with the Ago1x and RL-5BoxB sequences between the *Not*I and *Eco*RI sites. For overexpression of the full-length Ago1x, the canonical stop codon (TGA) was mutated to serine-encoding codon, TCA, in all backbones. The pri-let-7a sequence was cloned in the pcDNA3.1 plasmid between the *Hind*III and *Xba*I sites for overexpression in HeLa cells.

Dual-luciferase constructs were made in the pcDNA3.1 backbone. Nucleotide sequence encoding Renilla luciferase was cloned between the *Hind*III and *Bam*HI sites. Firefly luciferase coding sequence lacking the start codon, along with a linker sequence (as described before), was cloned between the *Xho*I and *Apa*I sites. The ISR of *AGO1* was cloned between the Renilla luciferase and the firefly luciferase coding sequences in the *Bam*HI and *Xho*I sites such that both luciferase coding sequences are in-frame.

To test the induction of the readthrough by tethering Ago1 and Ago1x downstream of the *AGO1* stop codon, two BoxB element sequences were cloned between the *AGO1* and the *FLuc* in place of *AGO1* ISR (see Fig 4C for a schematic). The *Bam*HI and *Xho*I sites were used for cloning. The BoxB sequence was PCR-amplified from the RL-5BoxB construct.

## Antibodies

Polyclonal anti-Ago1x antibody was generated in rabbit against the unique peptide sequence, RQNAVTSLDRRKLSKP, and was affinity-purified (Pierce Biotechnology). To confirm the specificity, the anti-Ago1x antibody was incubated with the same peptide at a final concentration of 1 μg/ml for 2 h with gentle agitation at room temperature before using it for Western blot (described below). Anti-Ago1 antibody (Cell Signaling Technologies, 9388; Sigma, SAB4200065), anti-Dicer (Sigma, SAB4200087), anti-HA (Sigma, clone 3F10, 11867423001; and Cell Signaling Technology, clone 6E2, 2367), anti-GW182 (Bethyl Laboratories, A302-329A), anti-puromycin (Merck Millipore, clone 12D10, MABE343), anti-Myc (Cell Signaling Technologies, 22725), anti-FLAG (Sigma, clone M2, F1804), anti-Actin (Sigma, A3854), anti-GAPDH (Sigma, G9295), and stabilized peroxidase-conjugated secondary antibodies (Thermo Fisher Scientific) were used at the manufacturer-recommended dilutions. Alexa Fluor-conjugated secondary antibodies (Molecular Probes) were used for immunofluorescence.

## SDS–PAGE and Western blot

Cells were lysed in the 1× cell lysis buffer (20 mM Tris, 150 mM NaCl, 1 mM EDTA, and 1% Triton X-100). For preparation of tissue lysates, tissues isolated from mice were washed in 1xPBS and lysed in RIPA buffer using a motor-driven tissue grinder. Cell or tissue lysates were electrophoresed for 1–1.5 h in a 7.5% (for Dicer and Ago proteins) or 10% (for others) SDS-polyacrylamide gel. To resolve Ago1x and Ago1 on the same gel, the 10% SDS-polyacrylamide gel was used and samples were electrophoresed for 3.5 h. Gels were soaked in the transfer buffer, and proteins were transferred onto the PVDF membrane (Immobilon-P; Merck Millipore) using the Trans-Blot semi-dry transfer apparatus (Bio-Rad Laboratories). Transferred membrane was blocked and treated with the primary antibody followed by the peroxidase-conjugated secondary antibody, as per the manufacturer's instructions. The blot was developed using the Clarity ECL reagent (Bio-Rad Laboratories), and the images were captured using a LAS-3000 imager (Fujifilm). Intensities of the bands were quantified using ImageJ.

## Luciferase-based translational readthrough assay

HEK293 cells were transfected with 500 ng/well of plasmid constructs expressing luciferase using Lipofectamine 2000 (Thermo Fisher Scientific) according to the manufacturer's protocol, at around 75–90% confluency in a 24-well plate. A plasmid expressing Renilla luciferase was co-transfected (100 ng/well), serving as the transfection control. Cells were lysed 24 h post-transfection, and luciferase activity was measured using Dual-Luciferase Reporter Assay System (Promega Corporation) in the GloMax Explorer System (Promega Corporation). For the readthrough assay in the presence of miRNA inhibitors, HeLa cells were transfected with 10 nM of let-7a inhibitor or control inhibitor (Sigma) along with the luciferase constructs. The cells were lysed 48 h post-transfection, and the luciferase activity was measured as described above. In all the constructs (including mutations and deletions), *FLuc* coding sequence was in-frame with *AGO1* (or *RHOA*) coding sequence.

## *In vitro* transcription and translation

Two μg of *Not*I-linearized plasmid DNA was transcribed *in vitro* using T7 RNA polymerase (Thermo Fisher Scientific) according to the manufacturer's protocol. RNA was purified using a GeneJET RNA purification kit (Thermo Fisher Scientific). The concentration and quality of the RNA were measured using BioPhotometer (Eppendorf). Four μg of RNA was subjected to *in vitro* translation using the Rabbit Reticulocyte Lysate System (Promega Corporation) according to the manufacturer's instructions. Luciferase activity was then measured using the Luciferase Assay System (Promega Corporation) in the GloMax Explorer System (Promega Corporation).

## Myc/His-tag-based readthrough assay

HEK293 cells overexpressing the *AGO1*-TGA-ISR-*Myc/His* construct were lysed, and the lysate was incubated with Ni-NTA agarose beads (Thermo Fisher Scientific) overnight at 4°C with tumbling. Beads were washed three times with wash buffer. Bound protein was eluted by 250 mM imidazole. Purified protein samples were run on a 12% SDS-polyacrylamide gel and detected with the anti-Ago1x and anti-Myc antibodies.

## Ribosome profiling data analysis

The mouse and human ribosome profiling datasets were obtained from NCBI's Sequence Read Archive (SRA) using the Aspera Connect facility provided by SRA in the SRA Toolkit. The datasets were then subjected to a BLAST search against the mRNA sequence of *AGO1* [transcript variant 1 of mouse (NM_153403.3) or human (NM_012199.4)] using the command line facility of the BLAST+ software. The BLAST results obtained for each dataset were stored separately according to the corresponding SRR numbers in a tab-delimited format. The rest of the analysis was done using in-house scripts in Python 3.6.0. Alignments with 100% identity (0 mismatches or gaps) and of length ≥ 24 were taken as *AGO1*-specific ribosomal footprint reads. These reads were then assigned to different regions of *AGO1* mRNA (5′UTR, coding sequence, inter-stop codon region (ISR), and 3′UTR) based on the P site, which is estimated as the central nucleotide position of the ribosomal footprint read. To avoid potential variability caused by the start, the canonical, and the in-frame stop codon peaks, reads with P sites aligning to the first 12 nucleotides after the start codon were not considered for the analyses. Similarly, reads with P sites aligning to the first 12 nucleotides after the canonical and the in-frame stop codon were also not considered for the analyses. However, they were included for graphical representation shown in Fig 2G and Appendix Fig S1. The reads with P sites aligning to the canonical stop codon were considered to be a part of the coding region, while those aligning to the downstream in-frame stop codon were considered for the inter-stop codon region.

Ribosome profiling datasets satisfying the following criteria [modified from Dunn *et al*, 2013] were taken as positive for translational readthrough of *AGO1*:

1   A minimum of 45 reads in the coding sequence.
2   At least 1 read in the second half of the inter-stop codon region.
3   At least 10% of the inter-stop codon region is covered by ribosomal footprints.
4   At least a threefold higher ribosomal footprint density in the inter-stop codon region compared to that in the rest of the 3′UTR. Ribosomal footprint density was calculated as the number of reads in the given region divided by the length of the given region.

The results generated from the above procedure were further verified using the online version of BLAST against the SRA databases using the corresponding SRX accession numbers with the following settings: The program selection was set to megablast, the word size was set to 24, and all the other parameters were left at default values. All the obtained reads were found to be the same as those identified by the above procedure, with the exception of some reads that either aligned with the negative strand or had gaps and mismatches (which have been excluded from the results).

## Analysis of 3-nucleotide periodicity

The ribosome profiling (Ribo-seq) and the mRNA-seq datasets of the U2-OS cell line were taken from the Elkon *et al* (2015) study (GEO accession no. GSE66927). The human transcriptome was downloaded from RefSeq. Adapter sequences were first removed from the FASTQ files using the Fastp tool. The resulting files were then subjected to alignment with the transcriptome using Bowtie 2. Only those reads (RNA-seq and Ribo-seq) that showed 100% match with the reference sequence

were considered. Ribo-seq files that did not show reads in the *AGO1* ISR were not included. To test if the Ribo-seq data exhibit 3-nucleotide periodicity, all reads aligning to the region near the start codon (−24 to 62; A of ATG being 0) were pooled for all genes for a particular file. Log values of number of reads +1 around the start codon were plotted as a heat map (Fig EV2B). As clear 3-nucleotide periodicity was seen for the fragment lengths of 27–29, only fragments of these lengths were considered for the analysis. After confirming 3-nucleotide periodicity, the proportions of reads in three frames that aligned to coding sequence, inter-stop codon region, and the 3′UTR of *AGO1* were calculated. The read proportions obtained from Ribo-seq data were compared with those obtained from mRNA-seq data of the same experiment.

## Mass spectrometry data analysis

Mass spectrometry raw files for mouse brain, myotubes, and liver tissues were downloaded using the Aspera Connect facility from the ProteomeXchange Consortium and were analyzed using MaxQuant version 1.6.2.6. The dataset was searched against a fasta file containing the whole mouse proteome (downloaded from the UniProt) and 20 possible Ago1x protein sequences (20 different amino acids in place of canonical stop codon). The number of missed cleavages was set to 5 since there are several R and K residues in the peptide sequence encoded by the ISR of *AGO1* (Fig 1A). The false discovery rate (FDR) for peptide spectrum matches was set to 0.05. Protein FDR filter was disabled. Default values were used for all other parameters. In-house Python scripts were used to identify peptides aligning to the canonical coding sequence and inter-stop codon region of *AGO1*. Using the "tblastn" tool, we made sure that the identified peptides are unique to the coding sequence and ISR of *AGO1*.

## N-peptide-mediated tethering and translational repression assay

HeLa cells were co-transfected with the following plasmids in a 24-well plate: (i) plasmids expressing N-HA-tagged or untagged Argonaute proteins, (ii) Renilla luciferase reporter construct RL-5BoxB (for luciferase activity) or FLAG-HA-RL-5BoxB (for Western blot), and (iii) pGL3 as a transfection control. Lipofectamine 2000 was used for transfection. Forty-eight hours post-transfection, cells were lysed and luciferase activity was measured using the Dual-Luciferase Reporter Assay System (Promega Corporation) in the GloMax Explorer System (Promega Corporation). Conditions were optimized to achieve more than 75% transfection efficiency using a construct that expresses green fluorescent protein. To achieve comparable expression, a four times more N-HA-Ago1x-expressing plasmid was transfected than N-HA-Ago1-expressing plasmid.

Plasmid expressing N-HA-Ago2ISR[Ago1] was constructed by replacing the *AGO1* sequence from the N-HA backbone with the *AGO2* coding sequence. ISR of *AGO1* was cloned downstream of the *AGO2* coding sequence between the *Xba*I and *Apa*I sites. The canonical stop codon of *AGO2*, TGA, was mutated to the serine codon, TCA, to obtain the overexpression of the full-length N-HA-Ago2ISR[Ago1] chimeric protein.

## RT–PCR analysis

Total RNA was isolated from cells using TRI reagent (Sigma) according to the manufacturer's instructions. The concentration and

quality of the RNA were measured using BioPhotometer (Eppendorf). Two to four µg of total RNA was reverse-transcribed using RevertAid Reverse Transcriptase (Thermo Fisher Scientific). For semi-quantitative RT–PCR, PCR amplification was performed for 25 cycles using gene-specific primers. Quantitative real-time PCR was done using SYBR Green mix (Takara Bio Inc.) on the iQ5 instrument (Bio-Rad Laboratories).

To study the Ago1x–mRNA interaction, FLAG-HA-tagged Ago1 and Ago1x were immunoprecipitated from HeLa cells and the associated RNA was extracted from the beads using the TRI reagent (Sigma). An equal volume of isolated RNA from beads as well as input samples was taken for cDNA synthesis using RevertAid Reverse Transcriptase (Thermo Fisher Scientific), and RT–PCR was performed as described above.

## Immunoprecipitation

Immunoprecipitation was done using anti-FLAG M2 Affinity Gel (Sigma, A2220). Cell lysate from HeLa cells overexpressing the FLAG-HA-tagged Ago protein was incubated with resin beads overnight with tumbling at 4°C. The beads were spun down and washed thrice. Immunoprecipitated proteins were extracted in the Laemmli buffer. Samples were then subjected to Western blotting using specific antibody. Ago1- or Ago1x-associated RNA was extracted using the TRI reagent (Sigma) according to the manufacturer's instructions. The concentration and the quality of the RNA were measured using BioPhotometer (Eppendorf).

Anti-GW182 antibody (Bethyl Laboratories, A302-329A) was used for immunoprecipitation of the endogenous GW182. The antibody was added to the pre-cleared lysate and incubated overnight with tumbling at 4°C, after which protein A-Sepharose beads were added and incubated for 4 more hours. The beads were spun down and washed twice. Immunoprecipitated proteins were extracted in the Laemmli buffer and subjected to Western blotting.

## Assays for miRNA–Ago interaction

FLAG-HA-tagged Ago1 and Ago1x were expressed in HeLa cells. Immunoprecipitation was done using anti-FLAG M2 Affinity Gel (Sigma, A2220). The miRNeasy Mini Kit (Qiagen) was used to isolate miRNAs associated with FLAG-HA-Ago1 and FLAG-HA-Ago1x, according to the manufacturer's protocol. The concentration and quality of the RNA were measured using BioPhotometer (Eppendorf). Immunoprecipitation performed on lysate from untransfected cells did not yield significant RNA and was excluded from further analysis. Isolated miRNAs were used for small RNA-seq and for qPCR using specific TaqMan probes.

Small RNA-seq was done by Clevergene Biocorp Pvt Ltd (Bengaluru, India). Adapter sequences, low-quality bases, and reads with lengths < 17 nucleotides and > 36 nucleotides were removed from the data. mRNA fragments and other non-coding RNA such as rRNA, tRNA, and snRNA were also removed. The remaining reads were mapped onto indexed human reference genome (GRCh38.p7) using mapper.pl script of miRDeep2. Reads mapped to reference genome were used to identity miRNA with miRDeep2 using known and novel miRNA identification parameters. Human and mouse mature miRNAs from miRBase 21 were used for miRNA prediction. miRNAs with miRDeep score < 1 were excluded from further

analysis. miRNAs associated with Ago1 and Ago1x are listed in Appendix Tables S1–S3.

## Confocal microscopy

Cells cultured on glass coverslips were fixed with 4% paraformaldehyde for 15 min at room temperature, washed with PBS, and incubated in blocking buffer (2% BSA, 0.1% saponin in PBS) at room temperature for 1 h. The cells were incubated overnight at 4°C with appropriate dilution of the primary antibody in blocking buffer. This was followed by three washes with PBS and 1 h of incubation in secondary antibody at room temperature. The cells were washed with PBS, and the coverslip was mounted on a slide using ProLong™ Gold Antifade Mountant (Thermo Fisher Scientific). Images were captured in LSM880 Confocal Laser Scanning Microscope (Zeiss).

## Proximity ligation assay (PLA)

Cells were prepared in the same way as described above for immunofluorescence. They were incubated overnight at 4°C with appropriate dilution of the primary antibodies in blocking buffer. The Duolink® *In Situ* Red Starter Kit Mouse/Rabbit (Sigma-Aldrich) was used as per the manufacturer's instructions. After washing the cells, PLA probes (1:5 dilution) were added to the cells and incubated at 37°C for 1 h. The cells were washed and ligation reaction was performed at 37°C for 30 min. After washing twice, the amplification solution was added to the cells and incubated for 100 min at 37°C. The cells were washed again and mounted on slides using Duolink® *In Situ* Mounting Medium with DAPI. Cells were then imaged using an Olympus FLUOVIEW FV3000 confocal microscope. Brightness and contrast were adjusted equally when required to visualize distinct dots. Multiple microscopic fields were analyzed using the "Analyze Particles" tool of ImageJ to quantify dots per cell.

## siRNAs, shRNAs, and miRNA inhibitors

HEK293 cells were transfected with *AGO1*-specific siRNAs (Flexi-Tube siRNAs from Qiagen, GS26523) using Lipofectamine 2000. The following shRNA constructs were used to generate stable knockdown of the respective genes in HeLa cells (Sigma MISSION® shRNA Plasmid DNA): for *AGO1*, TRCN0000007859 (which targets the distal 3′UTR); and for *GW182*, TRCN0000369459. Cells were transfected with shRNA constructs using lipofectamine 2000 and selected using 1 µg/ml of puromycin. MISSION Synthetic miRNA Inhibitors from Sigma were used to inhibit let-7a function. 10 nm of let-7a inhibitor (Sigma, HSTUD0001) and control inhibitor (Sigma, NCSTUD001) was transfected in HeLa cells using Lipofectamine 2000 (Thermo Fisher Scientific). Cells were lysed 48 h post-transfection for Western blotting or luciferase-based readthrough assay.

## Ribopuromycylation

HeLa cells were transfected with FLAG-HA-Ago1, FLAG-HA-Ago1x, or empty vector in six-well plates at 70–80% confluency. Transfection conditions were optimized to achieve > 75% efficiency using a plasmid expressing green fluorescent protein. Cells were treated with 91 µM puromycin (Sigma) for 3 h at 22 h post-transfection. Cells were then lysed and 50 µg of cell lysate was run on a 10%

SDS gel under denaturing conditions. Proteins were transferred onto a PVDF membrane. Western blotting was performed using anti-puromycin antibody (Merck Millipore, MABE343, clone 12D10). Protein synthesis was estimated by calculating band intensity using ImageJ. Transferred PVDF membranes were stained with Ponceau S to confirm equal loading and for normalization. Robust expression of FLAG-HA-tagged Ago1 and Ago1x was confirmed by Western blotting in every experiment.

**Metabolic labeling with [³⁵S]-methionine**

HeLa cells were transfected with FLAG-HA-Ago1 and FLAG-HA-Ago1x constructs. The medium was replaced with methionine-free serum-free RPMI 1640 medium containing 0.1 mCi/ml of [³⁵S]-methionine (American Radiolabeled Chemicals, Inc., ARS 0119) for 30 min at 22 h post-transfection. Cell lysate was electrophoresed on a 10% SDS-polyacrylamide gel. The gel was dried and exposed to Fujifilm BAS cassette 2025. The film was visualized using the Phosphorimager (Typhoon FLA 9000; GE).

**Statistics**

Two-sided Student's *t*-test was performed to test the significance of differences in experiments where samples showed normal distribution. Welch's correction was applied whenever equal variance was not observed. Non-parametric Mann–Whitney test was used when samples did not show normal distribution. Paired *t*-test was performed to compare endogenous levels of Ago1 and Ago1x after let-7a overexpression or inhibition. Linear regression analysis for miRNA enrichment was performed using GraphPad Prism 5.

# Data availability

The small RNA-seq data are available in Gene Expression Omnibus (GEO) repository of NCBI (accession number: GSE130608, https://www.ncbi.nlm.nih.gov/geo/query/acc.cgi?acc = GSE130608).

**Expanded View** for this article is available online.

## Acknowledgements

We thank Prof. Umesh Varshney for his valuable suggestions. We thank Prof. G Mugesh, Prof. Saumitra Das, Dr. Suvendra Bhattacharyya, and Dr. Sachin Kotak for sharing reagents. We thank the confocal microscope facility and the flow cytometry facility of Indian Institute of Science. This work was supported by the Wellcome Trust/DBT India Alliance Fellowship (IA/I/15/1/501833) awarded to SME. Authors gratefully acknowledge the financial support from the Director of the Indian Institute of Science (Part (2A) XII Plan (506/BC)), Department of Biotechnology (DBT) - Indian Institute of Science (IISc) Partnership Program for Advanced Research in Biological Sciences and UGC (University Grants Commission), India. SME is a recipient of Start-up Grant for Young Scientists from the Department of Science and Technology (DST)-Science and Engineering Research Board (SERB), India (YSS/2015/000989). AS and LEM are recipients of research fellowship from the Council of Scientific and Industrial Research (CSIR)-UGC, India. SS is a recipient of the KVPY (Kishore Vaigyanik Protsahan Yojana) fellowship from DST, India. PK is a recipient of a postdoctoral fellowship [SR/WOS-A/LS-1204/2015 (G)] under the Women Scientist Scheme from DST, India.

## Author contributions

SME and AS conceived the project and designed the experiments; AS, LEM, PK, AD, HRS, and SME conducted the experiments and analyzed the data; SS analyzed ribosome profiling and mass spectrometry data; PLF provided some reagents and helped in designing the experiments and data analysis; SME acquired funds and supervised the project; and SME and AS wrote the manuscript with inputs from all authors.

## Conflict of interest

The authors declare that they have no conflict of interest.

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
