## [Review Process File · The EMBO Journal]

Let-7a-regulated translational readthrough of mammalian AGO1 generates a microRNA pathway inhibitor

Anumeha Singh, Lekha E Manjunath, Pradipta Kundu, Sarthak Sahoo, Arpan Das, Harikumar R Suma, Paul L Fox and Sandeep M Eswarappa

Review timeline:	Submission date:	18th Sep 2018
	Editorial Decision:	30th Oct 2018
	Revision received:	25th Apr 2019
	Editorial Decision:	7th Jun 2019
	Revision received:	10th Jun 2019
	Accepted:	14th Jun 2019

Editor: Anne Nielsen/Ieva Gailite

Transaction Report:

1st Editorial Decision

30th Oct 2018

Thank you for submitting your manuscript for consideration by The EMBO Journal. It has now been seen by three referees whose comments are shown below.

As you will see from the reports, the three referees all express interest in the findings reported in your manuscript but also raise a number of concerns that you will have to address before they can support publication in The EMBO Journal. In particular, all three referees find that additional data is needed to understand how let-7a binding triggers translational readthrough of Ago1. They also point to a number of control experiments and further data analysis/description that should help increase the overall conclusiveness of the study. In my view, these points are all constructive and important and I would ask that you follow the recommendations from the referees in revising your study.

Given the referees' overall positive recommendations, I would like to invite you to submit a revised version of the manuscript, addressing the comments of all three reviewers. I should add that it is EMBO Journal policy to allow only a single round of revision, and acceptance of your manuscript will therefore depend on the completeness of your responses in this revised version.

REFeree REPORTS:

Referee #1:

In their manuscript "Let-7a-regulated translational readthrough of mammalian AGO1 generates a microRNA pathway inhibitor", the Eswarappa group builds upon a previous genome-wide screen by Eswarappa et al. (2014) that identified AGO1 as one of the read-through candidates. Here, Singh et al. verify programmed translational read-through of AGO1 and characterize its regulation by a let-7A binding site close to the STOP codon. The resulting c-terminally extended AGO1 protein variant, termed 'AGO1x', interacts with miRNAs and engages with target mRNAs, but fails to

recruit the GW182 and the mRNA degradation machinery. Thus, programmed translational read-through of AGO1 results in a dominant negative variant and global competition with miRNA mediated mRNA decay. Experiments are well designed and conducted and provide convincing evidence for a dominant negative AGO1x protein that is generated by translational read-through. I believe that some extension of their analysis on the regulatory potential of a nearby miRNA binding site (comment #1) could add important information and might uncover an extended feed-forward mechanism of post-transcriptional gene regulation. This work by the Eswarappa group uncovers an important regulation of global miRNA-mediated gene silencing and provides further support for the importance of programmed translational read-through. I recommend this manuscript for publication with minor revision.

Comment #1

It would be interesting to evaluate, if the regulatory potential of a nearby miRNA binding site on the AGO1 mRNA is mediated by binding of a wild-type RNA silencing complex or by an AGO1 read-through product itself (AGO1x). The authors could distinguish between these two hypotheses by tethering either wt AGO1 or AGO1x downstream of the STOP codon of AGO1 mRNA. This would be particularly interesting to evaluate, because the authors do not observe a canonical miRNA effect (RNA decay) when evaluating this let7a site. If the observed effect is mediated through AGO1x, the authors might have uncovered a feed-forward regulatory mechanism that could extend to other miRNA targets with binding sites close to the STOP codon (which could be tested globally in a future study).

Minor point:

Page 15: "...miRNAs are also reported to regulate transcription in the nucleus (Liu, Lei et al., 2018)..." to complete the reference on diverse functions of miRNAs, the authors should also cite the recent report by Sarshad et al. (Sarshad, A. A. et al. Argonaute-miRNA Complexes Silence Target mRNAs in the Nucleus of Mammalian Stem Cells. *Molecular Cell* 71, 1040-1050.e8 (2018)) describing post-transcriptional regulation by nuclear AGO during that expands the miRNA-target space in specific cell types during development.

Referee #2:

In this manuscript Singh et al. reveal that Ago1 undergoes a relatively very efficient stop codon readthrough (SC-RT) to produce a C-terminally extended isoform Ago1x. The mechanism of this so-called programmed SC-RT has not been studied in detail; the authors just show - quite convincingly, though - that binding of the let-7a microRNA to the 5' end of the extension sequence is required and speculate but not show that it could serve as a "roadblock" for the terminating ribosome to somehow stimulate SC-RT. Furthermore, they claim that the Ago1x isoform can load numerous miRNAs on target mRNAs, like Ago1, however, that it at the same time fails to bind GW182 which prevents it from repressing their translation. As such, the authors propose that the naturally generated Ago1x isoform acts as a global miRNA pathway inhibitor in many tissues.

This is a clearly written, technically well executed, fairly interesting story supported by numerous experiments that, in my opinion, has a merit to be eventually published in the EMBO J. However, there are several issues that should be addressed before this paper will be suitable for publication.

Major criticism:

- 1) The let-7a part of the story is very well done but to convince me that let-7a is really directly involved in stalling etc. of the terminating ribosomes to promote SC-RT, I would like to see what happens if the let-7a binding region has been moved by 10 or 20 nt downstream. Moreover, since your retics data (Fig. 1E) suggest that at least 57 nt are required to detect efficient RT (36 nt - containing the spacer plus let-7a binding region - are not enough), I was wondering that it is probably a lot more complex than it may seem.
- 2) The chapter: "Ago1x can load miRNAs on target mRNAs"; all experiments presented here and in the following chapter were carried out with overexpressed proteins on top of the endogenous Ago1. I am not very familiar with the protein-protein interactions within the RISC complex, or with all known Ago1 interacting partners, but can you rule out that your Ago1x IP samples do not contain

any endogenous Ago1? (Its presence could be achieved by bridging interactions....) I think to be 100% sure that what you detect in your IP samples does come down merely via the Ago1x-mediated interactions, these experiments should be repeated in the endogenous Ago1 knock-down (or knock-out).

3) Fig. 4E and F; on a similar note, this particular Western blot (IB, anti-GW 182) looks the least convincing (smudgy) of all (by far) and I am not convinced that it supports the author's claims ("vector control" is missing, "untransfected" control is shown instead (why?) but is missing in the "Input" section?). What if GW182 is a low abundant protein fully trapped by endogenous Ago1 (with a slightly higher affinity for it than Ago1x might have) in a functional complex and that's why you find it neither in the Ago1x IP sample nor co-localizing in cells? In any case, since this is - in my mind - the key experiment that you use to build your model upon, you should try to repeat it in the Ago1 knock-down or knock-out, try to IP GW182 looking for Ago1x, as well as employ some direct in vitro protein-protein binding assays to check if these two proteins really do not bind.

4) I was not really impressed by the RiboPuro assay - I do not think it is sensitive enough to entitle you to claim that Ago1x is a global miRNA pathway inhibitor. Besides, I would expect a decrease and not a slight increase in global translation with Ago1 overexpressed...? What if you try for example pulse-chase metabolic labeling with hot leucine? In the chase phase you could see that hot Leu incorporation in your Ago1x sample declines a lot slower than in your Ago1 sample, if my logic is correct.

Minor comments:

1) Page 4 - Introduction; perhaps it would be helpful to describe how Ago1 and GW182 interact, both physically and functionally.

2) Page 6 - Fig. 1A ; you talk about 99 nt but Fig. 1A shows aa residues.

3) Fig. 1D; it is no surprise that UGA showed the highest SC-RT of all stop codons; it is considered to be the leakiest of all three stops. You should acknowledge that ... see for ex. PMID: 26176195.

4) Page 8 - "Ago1x is 34 (33 encoded by the ISR and 1 by the canonical stop codon) amino acids..."; I would reverse the order: "(1 encoded by the canonical stop codon and 33 by the ISR)"

5) Page 8 - "The intensity of the ~100 kDa band was reduced in the lysates of these cells showing that the band indeed represents an isoform of Ago1 (Fig 2D)."; unclear, please explain.

6) Fig. 2F; the brain tissue seem to carry two major forms of Ago1x. Any idea what it could mean?

7) Ribosome profiling data do not speak for a very efficient SC-RT; they definitely do not support the calculated ratio between Ago1 vs. Ago1x given in Fig. 2E (60/40%). Taking into account ~10-30% SC-RT efficiency obtained in various reporter assays, I would guess that this ratio is unreal and, in reality, is a lot smaller in disfavor of Ago1x.

8) Page 10; as far as I know, the terminating ribosome with eRFs bound protects ~28 nt; therefore, at least 9-10 nt past the stop codon should be protected and not just 6. Regardless, the let-7a binding region should still be free. Please correct if agree.

9) Page 10 - let-7a inhibitor; please explain briefly how it works.

10) Page 11; as far as I know, UGA is preferentially decoded by either Trp or Cys-tRNAs - never heard of Ser-tRNA; please see PMID:25056309, 26759455, 25733896.

11) Have you searched for nucleotide and protein sequences homologous to the Ago1 ISR region in the mammalian genomes and proteomes, respectively?

12) Taking into account that Ago1, as well as Ago1x can load let-7a onto its own mRNA, what if the major role of this isoform is to regulate the expression of fully functional Ago1?

I thank you for the opportunity to review this article. Leos Shivaya Valasek

Referee #3:

In this manuscript Singh and coworkers describe translational read-through of mammalian AGO1 generating a protein variant that functions a microRNA inhibitor. The author present evidence that let-7 binding to a conserved region in the 3'UTR downstream of the annotated stop codon results in read-through. The extended variant of AGO1 (AGO1x) does not interact with GW180 proteins and is unable to function in translational repression.

The manuscript is quite interesting describing a novel negative regulator of miRNA function. However, before considering the manuscript for publication some additional experiments and analyses are needed to support the conclusions

1. Despite the generation of an antibody for the read-through peptide, it is recommended to generate a full length AGO1 expression construct with a N-terminal epitope tag (e.g HA) and a C-terminal tag (e.g. myc-tag) right after the ISR sequence. This way the amount of AGO1 and AGO1x can be better and more accurately estimated than with the AGO1x specific antibody.

2. The presentation of the Ribo-seq evidence is in its current form no convincing, since any information of the frame of the Ribo-seq reads is not presented. The ISR region should be examined for 3nt periodicity (by using RiboTaper or similar tools).

3. Likewise the analyses of mass spec of deep proteomes needs to be expanded. The authors should more thoroughly examine deep proteome data for the expected tryptic peptide. MS/MS spectra from the different proteome data sets (if raw data is available is submitted) should be presented and the information how far above the threshold the score for this peptide is in the respective data sets. Ideally, one would like to see that that the respective peptide is synthesized and detected in different biological samples by targeted mass spectrometry.

4. The authors show that let-7 binding is involved in the read-through. Singh and colleagues should mutate the double tagged construct in a way to prevent microRNA binding without changing the ISR peptide sequence to rule out that the peptide sequence is involved. Furthermore, a microRNA mimic can be designed that binds to the mutated sequence and should rescue read-through. Such results would indicate that the read-through is not dependent on a specific microRNA.

Minor issue:

Why is there a linker sequence between ISR and the AUG-less Fluc reading frame. Why not fusing ISR to the Fluc CDS.

1st Revision - authors' response

25th Apr 2019

Response to Reviewers' comments

(Reviewers' comments in bold, our responses in italics):

Referee#1:

In their manuscript "Let-7a-regulated translational readthrough of mammalian AGO1 generates a microRNA pathway inhibitor", the Eswarappa group builds upon a previous genome-wide screen by Eswarappa et al. (2014) that identified AGO1 as one of the read-through candidates. Here, Singh et al. verify programmed translational read-through of AGO1 and characterize its regulation by a let-7A binding site close to the STOP codon. The resulting c-terminally extended AGO1 protein variant, termed 'AGO1x', interacts with miRNAs and engages with target mRNAs, but fails to recruit the GW182 and the mRNA degradation machinery. Thus, programmed translational read-through of AGO1 results in a dominant negative variant and global competition with miRNA mediated mRNA decay.

Experiments are well designed and conducted and provide convincing evidence for a dominant negative AGO1x protein that is generated by translational read-through. I believe that some extension of their analysis on the regulatory potential of a nearby miRNA binding

site (comment #1) could add important information and might uncover an extended feed-forward mechanism of post-transcriptional gene regulation. This work by the Eswarappa group uncovers an important regulation of global miRNA-mediated gene silencing and provides further support for the importance of programmed translational read-through. I recommend this manuscript for publication with minor revision.

We thank the Reviewer for the kind words in appreciation of our work. We have conducted the experiment suggested by this Reviewer.

Comment#1

It would be interesting to evaluate, if the regulatory potential of a nearby miRNA binding site on the AGO1 mRNA is mediated by binding of a wild-type RNA silencing complex or by an AGO1 read-through product itself (AGO1x). The authors could distinguish between these two hypotheses by tethering either wt AGO1 or AGO1x downstream of the STOP codon of AGO1 mRNA. This would be particularly interesting to evaluate, because the authors do not observe a canonical miRNA effect (RNA decay) when evaluating this let7a site. If the observed effect is mediated through AGO1x, the authors might have uncovered a feed-forward regulatory mechanism that could extend to other miRNA targets with binding sites close to the STOP codon (which could be tested globally in a future study).

Response: This is an excellent suggestion. We adopted BoxB-N-peptide tethering system to address this. We cloned BoxB element downstream to the AGO1 stop codon followed by luciferase coding sequence. There was no AGO1 ISR (Inter stop codon region) in the construct. BoxB element will bind N-peptide-tagged proteins. When this construct was co-transfected with N peptide-tagged Ago1 or Ago1x, there was a significant induction of readthrough as indicated by luciferase assay.

Induction by Ago1x was more than Ago1 (Fig 4C). This result is consistent with our other assays showing induction of AGO1 readthrough by Let-7a miRNA. As per the suggestion of this Reviewer, we are currently searching for other transcripts which have miRNA binding sites close to the stop codon.

Minor point:

Page 15: ..."miRNAs are also reported to regulate transcription in the nucleus (Liu, Lei et al., 2018)..." to complete the reference on diverse functions of miRNAs, the authors should also cite the recent report by Sarshad et al. (Sarshad, A. A. et al. Argonaute-miRNA Complexes Silence Target mRNAs in the Nucleus of Mammalian Stem Cells. Molecular Cell 71, 1040-1050.e8 (2018)) describing post-transcriptional regulation by nuclear AGO during that expands the miRNA-target space in specific cell types during development.

Response: Thanks for bringing this report to our attention. We have included this in the revised manuscript.

Referee #2:

In this manuscript Singh et al. reveal that Ago1 undergoes a relatively very efficient stop codon readthrough (SC-RT) to produce a C-terminally extended isoform Ago1x. The

mechanism of this so-called programmed SC-RT has not been studied in detail; the authors just show - quite convincingly, though - that binding of the let-7a microRNA to the 5' end of the extension sequence is required and speculate but not show that it could serve as a "roadblock" for the terminating ribosome to somehow stimulate SC-RT. Furthermore, they claim that the Ago1x isoform can load numerous miRNAs on target mRNAs, like Ago1, however, that it at the same time fails to bind GW182 which prevents it from repressing their translation. As such, the authors propose that the naturally generated Ago1x isoform acts as a global miRNA pathway inhibitor in many tissues.

This is a clearly written, technically well executed, fairly interesting story supported by numerous experiments that, in my opinion, has a merit to be eventually published in the EMBO J. However, there are several issues that should be addressed before this paper will be suitable for publication.

We thank the Reviewer for his kind words and valuable insights. We have addressed his concerns by performing additional experiments as described below.

Major criticism:

1) The let-7a part of the story is very well done but to convince me that let-7a is really directly involved in stalling etc. of the terminating ribosomes to promote SC-RT, I would like to see what happens if the let-7a binding region has been moved by 10 or 20 nt downstream. Moreover, since your retics data (Fig. 1E) suggest that at least 57 nt are required to detect efficient RT (36 nt - containing the spacer plus let-7a binding region - are not enough), I was wondering that it is probably a lot more complex that it may seem.

Response: As suggested by the Reviewer, we performed luciferase-based readthrough assay using a construct (AGO1 TGA disISR FLuc) where Let-7a binding site was moved by 18 nts away from the canonical stop codon. This construct showed reduced (about 4-fold) readthrough compared to the construct with wild-type ISR (Inter stop codon region) (Fig 3F). Furthermore, just tethering of N-peptide-tagged Ago proteins downstream to the stop codon could induce readthrough (Fig. 4C). We agree to the point that sequence surrounding Let-7a binding site also contributes to readthrough.

2) The chapter: "Ago1x can load miRNAs on target mRNAs"; all experiments presented here and in the following chapter were carried out with overexpressed proteins on top of the endogenous Ago1. I am not very familiar with the protein-protein interactions within the RISC complex, or with all known Ago1 interacting partners, but can you rule out that your Ago1x IP samples do not contain any endogenous Ago1? (Its presence could be achieved by bridging interactions....) I think to be 100% sure that what you detect in your IP samples does come down merely via the Ago1x-mediated interactions, these experiments should be repeated in the endogenous Ago1 knock-down (or knock-out).

Response: To address this concern we generated stable AGO1 knock down HeLa cells using a specific shRNA that targets 3'UTR of AGO1. These cells also showed interaction of Ago1x with

miRNAs, target mRNA, Dicer and did not show interaction with GW182 (Fig EV3). These results show that interaction of Ago1x with miRNA, Dicer and target mRNA is not via Ago1.

3) Fig. 4E and F; on a similar note, this particular Western blot (IB, anti-GW 182) looks the least convincing (smudgy) of all (by far) and I am not convinced that it supports the author's claims ("vector control" is missing, "untransfected" control is shown instead (why?) but is missing in the "Input" section?).

Response: We have repeated this experiment and provided new results with vector control. The present blot is not smudgy and we believe is more convincing (Fig 6A and its source file).

Furthermore, we confirmed the specificity of GW182 antibody (Bethyl laboratories, A302-329A) using GW182 knockdown cells which showed reduced expression of the band (i.e., GW182) recognized by this antibody (Fig EV5).

What if GW182 is a low abundant protein fully trapped by endogenous Ago1 (with a slightly higher affinity for it than Ago1x might have) in a functional complex and that's why you find it neither in the Ago1x IP sample nor co-localizing in cells? In any case, since this is - in my mind - the key experiment that you use to build your model upon, you should try to repeat it in the Ago1 knock-down or knock-out, try to IP GW182 looking for Ago1x, as well as employ some direct in vitro protein-protein binding assays to check if these two proteins really do not bind.

Response: We have addressed this concern by performing following additional experiments:

1. *As suggested, we repeated the IP in AGO1 knockdown cells. Similar to cells with AGO1 (i.e., wild-type), we did not observe GW182 interaction with Ago1x in AGO1 knockdown cells (Fig EV4A).*
2. *As suggested, we immunoprecipitated endogenous GW182. While Ago1 co-immunoprecipitated with GW182, Ago1x did not. This result also supports our conclusion that Ago1x, unlike Ago1, does not interact with GW182 (Fig EV4B).*
3. *To further confirm this, we performed Proximity Ligation Assay (PLA) which detects protein-protein interactions with high sensitivity and specificity; even weak and transient interactions can be detected by this method. This assay also showed that GW182 interacts with Ago1, but not with Ago1x (Fig 6C).*
4. *We next transferred the ISR of AGO1 to the 3' end of AGO2 such that C-terminally extended Ago2 isoform (Ago2ISR^{Ago1}, with a peptide encoded by the ISR of AGO1) is made. Like Ago1x, Ago2ISR^{Ago1} also lost its ability to repress translation (Fig 7D) and did not show interaction with GW182 in PLA (Fig EV4C). This result also shows that C-terminal extension in Ago1x encoded by the ISR of AGO1 interferes with the interaction between GW182 and Ago proteins carrying the C-terminal extension.*

Because of their high molecular weight (Ago1: 110 kDa and GW182: 210 kDa) purifying Ago1 and GW182 poses a technical challenge (well accepted in the field) and therefore, we could not perform the in vitro interaction experiment with purified proteins. Nonetheless, we have shown the lack of interaction between Ago1x and GW182 using three different methods –immunoprecipitation (both

ways, IP of Ago proteins and IP of GW182; and in both normal and AGO1-knock down cells), immunolocalization and the very sensitive and specific Proximity Ligation Assay. We believe that, together these results convincingly show that Ago1x does not interact with GW182 in HeLa cells.

4) I was not really impressed by the RiboPuro assay - I do not think it is sensitive enough to entitle you to claim that Ago1x is a global miRNA pathway inhibitor. Besides, I would expect a decrease and not a slight increase in global translation with Ago1 overexpressed...? What if you try for example pulse-chase metabolic labeling with hot leucine? In the chase phase you could see that hot Leu incorporation in your Ago1x sample declines a lot slower than in your Ago1 sample, if my logic is correct.

Response:

- (1) *To validate Ribopuromycylation (RPM) assay, we performed the experiment in GW182 knockdown cells where miRNA pathway is inhibited causing increased global translation. RPM was able to reveal this in the form of increased puromycin signal (Fig EV5).*
- (2) *To further confirm our results, we performed metabolic labeling using ³⁵S-methionine as described previously to study global translation (Ruoff R et al, 2016 PNAS, PMID: 27313212; Shenton D et al, 2006 J. Biol. Chem. PMID: 16849329). In consistence with RPM assay, we observed increased ³⁵S-methionine incorporation in cells overexpressing Ago1x (Fig 8B). Like RPM assay, we did not observe decreased translation in Ago1 overexpressing cells. The reason could be that Ago proteins are abundant and therefore are not limiting in the cell to execute miRNA-mediated translational repression.*

Minor comments:

1) Page 4 - Introduction; perhaps it would be helpful to describe how Ago1 and GW182 interact, both physically and functionally.

Response: We have included this information in the revised manuscript.

2) Page 6 - Fig. 1A ; you talk about 99 nt but Fig. 1A shows aa residues.

Response: Thanks for pointing out this mistake. We have modified the sentence in the revised manuscript.

3) Fig. 1D; it is no surprise that UGA showed the highest SC-RT of all stop codons; it is considered to be the leakiest of all three stops. You should acknowledge that ... see for ex. PMID: 26176195.

Response: We have acknowledged this point and cited the above reference in the revised manuscript.

4) Page 8 - "Ago1x is 34 (33 encoded by the ISR and 1 by the canonical stop codon) amino acids..."; I would reverse the order: "(1 encoded by the canonical stop codon and 33 by the ISR)"

Response: We have made this change in the revised manuscript.

5) Page 8 - "The intensity of the ~100 kDa band was reduced in the lysates of these cells showing that the band indeed represents an isoform of Ago1 (Fig 2D)."; unclear, please explain.

Response: We have rephrased the sentence to make it clear. In cells transfected with AGO1-specific siRNAs, the intensity of ~100 kDa band detected by anti-Ago1x antibody was reduced. This shows that the band indeed represents an isoform of AGO1.

6) Fig. 2F; the brain tissue seem to carry two major forms of Ago1x. Any idea what it could mean?

Response: At this point we can only speculate on this interesting observation. This could be a post-translationally modified protein which is reported previously for Ago proteins including Ago1 (e.g., Poly-ADP ribosylation, Leung et al., 2011 Mol Cell, PMID: 21596313).

7) Ribosome profiling data do not speak for a very efficient SC-RT; they definitely do not support the calculated ratio between Ago1 vs. Ago1x given in Fig. 2E (60/40%). Taking into account ~10-30% SC-RT efficiency obtained in various reporter assays, I would guess that this ratio is unreal and, in reality, is a lot smaller in disfavor of Ago1x.

Response: We thank the Reviewer for this insight. We agree with the point that proportion of Ago1x in cells does not indicate the efficiency of translational readthrough in AGO1 as it also depends on the stability of endogenous Ago1x. Therefore, it is not correct to estimate the % of readthrough by the proportion of readthrough product. Ribosome profiling data is also an inaccurate method to calculate % of readthrough. This is because higher density of ribosome foot-prints can be seen due to ribosome pausing as well as higher translation which cannot be distinguished by ribosome profiling.

8) Page 10; as far as I know, the terminating ribosome with eRFs bound protects ~28 nt; therefore, at least 9-10 nt past the stop codon should be protected and not just 6. Regardless, the let-7a binding region should still be free. Please correct if agree.

Response: We rechecked the following reference: Fig. 2B in Ingloia NT et al, Science 2009 PMID: 19213877. This paper clearly shows that the footprints of ribosomes extend till 9 nucleotides (including the stop codon) into the 3'UTR. i.e., 6 nucleotides after the stop codon.

9) Page 10 - let-7a inhibitor; please explain briefly how it works.

Response: miRNA inhibitors (from Sigma) are double-stranded small RNA molecules that bind specific miRNA and inhibit its function. We have included this information in the revised manuscript.

10) Page 11; as far as I know, UGA is preferentially decoded by either Trp or Cys-tRNAs - never heard of Ser-tRNA; please see PMID: 25056309, 26759455, 25733896.

Response: We agree that Trp and Cys are encoded in place of UGA stop codon. However, serine incorporation in UGA is also supported by previous studies (Chittum et al., 1998 Biochemistry PMID: 9692979; Eswarappa et al., 2014 Cell PMID: 24949972; Hatfield 1972 PNAS PMID: 4562751). In fact, Hatfield (1972) has demonstrated that seryl-tRNA can bind UGA stop codon.

11) Have you searched for nucleotide and protein sequences homologous to the Ago1 ISR region in the mammalian genomes and proteomes, respectively?

Response: Yes, we have done this by BLAST. We did not find any other mammalian sequence that is homologous to AGO1 ISR at nucleotide level or amino acid level.

12) Taking into account that Ago1, as well as Ago1x can load let-7a onto its own mRNA, what if the major role of this isoform is to regulate the expression of fully functional Ago1?

Response: Excellent point. Yes, it is another way of looking at this process. Readthrough not only reduces the functional isoform (Ago1), it also generates an inactive isoform (Ago1x). Both effects contribute to the negative regulation of miRNA pathway. At this point, we don't know whose contribution is more.

I thank you for the opportunity to review this article. Leos Shivaya Valasek

We thank you for your constructive suggestions.

Referee #3:

In this manuscript Singh and coworkers describe transitional read-through of mammalian AGO1 generating a protein variant that functions a microRNA inhibitor. The author present evidence that let-7 binding to a conserved region in the 3'UTR downstream of the annotated stop codon results in read-through. The extended variant of AGO1 (AGO1x) does not interact with GW180 proteins and is unable to function in translational repression.

The manuscript is quite interesting describing a novel negative regulator of miRNA function. However, before considering the manuscript for publication some additional experiments and analyses are needed to support the conclusions.

We thank the Reviewer for encouraging words and valuable suggestions. We have addressed the concerns by performing additional analyses and experiments as described below.

1. Despite the generation of an antibody for the read-through peptide, it is recommended to generate a full length AGO1 expression construct with a N-terminal epitope tag (e.g HA) and a C-terminal tag (e.g. myc-tag) right after the ISR sequence. This way the amount of AGO1 and AGO1x can be better and more accurately estimated than with the AGO1x specific antibody.

Response: As suggested here we made a construct with N-terminal renilla luciferase and C-terminal firefly luciferase construct in which AGO1 ISR was cloned between them. This construct showed \approx 7% readthrough in HeLa cells and \approx 9% in in vitro translation system. As shown in the Results we

have used multiple tagged constructs to estimate readthrough efficiency – myc-tag, luciferase-tag and GFP-tag. As we see in the Results, the efficiency of readthrough (relative amounts of Ago1 and Ago1x) in these overexpression constructs highly depends on the tag as well as the assay – it is about 20% in single luciferase-based assay, 30% in fluorescence-based assay, 7% in dual luciferase-based assay and about 5 % in Western blot. This shows that overexpressed and tagged Ago1x has different stability (or half-life) compared to endogenous Ago1x.

2. The presentation of the Ribo-seq evidence is in its current form no convincing, since any information of the frame of the Ribo-seq reads is not resented. The ISR region should be examined for 3nt periodicity (by using RiboTaper or similar tools).

Response: As suggested we performed 3-nt periodicity for the ribosome profiling data from U2-OS cells because 16 ribosome profiling data files from this cell line showed evidence of translational readthrough (> 20 fold increase in ribosomal density in the ISR compared to 3'UTR, Table EV1). Ribosome profiling data files from the study Elkon et. al. 2015 (PMID: 26538417; GEO no. GSE66927) done in U2-OS cells that showed positive results were pooled together and 3-nt periodicity analysis was done (details provided in Methods). The distribution of Ribo-seq reads in the ISR was non-uniform and it was similar to coding sequence with majority of them in 0th frame which is consistent with translational readthrough of AGO1 (Fig EV2B).

3. Likewise the analyses of mass spec of deep proteomes needs to be expanded. The authors should more thoroughly examine deep proteome data for the expected tryptic peptide. MS/MS spectra from the different proteome data sets (if raw data is available is submitted) should be presented

Response: We have provided MS/MS spectra of readthrough peptides found in three different mouse tissues in the revised manuscript (Appendix Fig S2).

and the information how far above the threshold the score for this peptide is in the respective data sets.

Response: We have provided the scores for all the peptides detected by our analysis in Table EV2 of revised manuscript. All of them had False Discovery Rate (FDR) less than < 0.05. Furthermore, a recent preprint uploaded in bioRxiv demonstrates the readthrough-specific peptide (QNAVTSLLDR) from breast cancer cells (MDA-MB-231) using mass-spectrometry further supporting our observation (<https://www.biorxiv.org/content/10.1101/603506v1>).

Ideally, one would like to see that that the respective peptide is synthesized and detected in different biological samples by targeted mass spectrometry.

Response: As per this suggestion, we analyzed deep proteomics data from multiple organs. Our analyses identified AGO1 readthrough peptide in deep proteomics data from brain (PMID: 26523646), mouse myotubes (PMID: 25616865) and liver (PMID: 25470552) derived from mouse (Table EV2). As mentioned above, AGO1 readthrough peptide has been identified in breast cancer cells. Interestingly, Ago1x-specific peptide (AVQVHQDTRLTM(ox)YFAYR) was detected

independently in five different mouse liver samples further increasing the confidence in the in vivo existence of Ago1x isoform.

4. The authors show that let-7 binding is involved in the read-through. Singh and colleagues should mutate the double tagged construct in a way to prevent microRNA binding without changing the ISR peptide sequence to rule out that the peptide sequence is involved.

Response: Peptide generated by the inter-stop codon region (ISR) is unlikely to influence readthrough as it is generated only after the readthrough. Nonetheless, the following experiment addresses this concern: We shifted the miRNA binding site in ISR 18 nucleotides away from the stop codon. This does not alter the peptide sequence encoded in that region. This construct exhibited reduced readthrough efficiency (~4 fold) showing that the location of miRNA binding site, but not the peptide derived from it, is important for readthrough (Figure 3F).

Furthermore, a microRNA mimic can be designed that binds to the mutated sequence and should rescue read-through. Such results would indicate that the read-through is not dependent on a specific microRNA.

Response: To show that the readthrough does not depend on a specific microRNA (i.e., Let 7a in case of AGO1), we replaced the inter-stop codon region (ISR) with boxB element. This element will bind N-peptide tagged Ago proteins. When this construct was transfected in cells expressing N-peptide-tagged Ago1 or Ago1x, there was an induction of readthrough (Figure 4C). This experiment shows that miRNA-independent interaction of Ago proteins with the ISR can also induce readthrough in AGO1. In other words, readthrough is not dependent on specific miRNA as rightly pointed out by this Reviewer. Furthermore, this result also supports the conclusion that ISR peptide sequence is not responsible for readthrough (see the comment above).

Minor issue:

Why is there a linker sequence between ISR and the AUG-less Fluc reading frame. Why not fusing ISR to the Fluc CDS.

Response: Flexible linker sequences minimize the interference between separate folding domains (Ago1 and Luciferase/GFP in our case).

Reference: Methods Mol Biol. 2011; 680: 29–43. PMID: 21153371.

2nd Editorial Decision

7th Jun 2019

Thank you for submitting a revised version of your manuscript. I have taken over the handling of your manuscript from my colleague Anne Nielsen, who has meanwhile left our office. The manuscript has now been seen by the two of original referees, who find that their main concerns have been addressed and are now broadly in favour of publication of the manuscript. There remain only a few editorial issues that have to be dealt with before I can extend formal acceptance of the manuscript.

REFeree REPORTS:

Referee #1:

The authors have sufficiently addressed all my concerns in the revised version.

Referee #2:

I have no further comments, am impressed by the thorough work that went into this revision and want to congratulate the authors on this wonderful achievement!

2nd Revision - authors' response

10th Jun 2019

The authors performed the requested editorial changes.

3rd Editorial Decision

14th Jun 2019

Editor accepted the revised manuscript.

Corresponding Author Name: Sandeep M Eswarappa

Journal Submitted to: The EMBO Journal

Manuscript Number: EMBOJ-2018-100727R